# The MOSAiC drift from October 2019 to July 2020: Sea ice conditions from space and comparison with previous years

Thomas Krumpen[1*], Luisa von Albedyll[1*], Helge F. Goessling[1*], Stefan Hendricks[1*], Bennet Juhls[1*], Gunnar Spreen[2*], Sascha Willmes[3*], H. Jakob Belter[1], Klaus Dethloff[1], Christian Haas[1], Lars Kaleschke[1], Christian Katlein[1], Xiangshan Tian Kunze[1], Robert Ricker[1], Philip Rostosky[2], Janna Rückert[2], Suman Singha[4], Julia Sokolova[5]

[*]These authors contributed equally to this work

[1]Alfred Wegener Institute, Helmholtz Centre for Polar and Marine Research, Am Handelshafen 12, 27570 Bremerhaven, Germany
[2]University of Bremen, Institute of Environmental Physics, Otto-Hahn Allee 1, 28359 Bremen, Germany
[3]University of Trier, Environmental Meteorology, Universitätsring 15, 54296 Trier, Germany
[4]German Aerospace Center, Remote Sensing Technology Institute, SAR Signal Processing, Am Fallturm 9, 28359 Bremen, Germany
[5]Arctic and Antarctic Research Institute, Ulitsa Beringa, 38, Saint Petersburg, 199397, Russia

*Correspondence to*: Thomas Krumpen (tkrumpen@awi.de)

**Abstract.**

We combine satellite data products to provide a first and general overview of the physical sea-ice conditions along the drift of the MOSAiC expedition (international Multidisciplinary drifting Observatory for the Study of Arctic Climate) and a comparison with previous years (2005/2006 – 2018/2019). We find that the MOSAiC drift was around 20% faster than the climatological mean drift, as a consequence of large-scale low-pressure anomalies prevailing around the Barents-Kara-Laptev Sea region between January and March. In winter (October - April), satellite observations show that the sea-ice in the vicinity of the Central Observatory (CO, 50 km radius) was rather thin compared to the previous years along the same trajectory. Unlike ice thickness, satellite-derived sea-ice concentration, lead frequency, and snow thickness during winter months were close to the long-term mean with little variability. With the onset of spring and decreasing distance to Fram Strait, variability in ice concentration and lead activity increased. In addition, frequency and strength of deformation events (divergence, convergence, and shear) were higher during summer than during winter. Overall, we find that sea-ice conditions observed within 5 km distance of the CO are representative for the wider (50 km and 100 km) surroundings. An exception is the ice thickness: Here we find that sea-ice within 50 km radius of the CO was thinner than sea-ice within a 100 km radius by a small but consistent factor (4%) for successive monthly averages. Moreover, satellite acquisitions indicate that the formation of large melt ponds began earlier on the MOSAiC floe than on neighbouring floes.

## 1 Introduction

In October 2019, the icebreaker *Polarstern* operated by the Alfred Wegener Institute Helmholtz Centre for Polar and Marine Research (AWI, 2017), was moored to an ice floe north of the Laptev Sea (October 4 at 85°N, 136°E). Scientists from 16 different nations onboard of *Polarstern* embarked on a one-year long journey along the Transpolar Drift towards Fram Strait (Fig. 1). The goal of the international Multidisciplinary drifting Observatory for the Study of Arctic Climate (MOSAiC) project is to better understand and quantify relevant processes within the coupled atmosphere-ice-ocean system and ecological and biogeochemical feedbacks, ultimately leading to much improved climate and Earth System models. The Central Observatory (CO) with comprehensive instrumentation was set up on an ice floe measuring roughly 2.8 km × 3.8 km. The floe was part of a loose assembly of pack ice, less than a year old, which had survived the 2019 summer melt season (Krumpen et al., 2020). Around the CO, a distributed network (DN) of autonomous buoys was installed in a 40 km radius on 55 additional residual ice floes of similar age (Krumpen and Sokolov, 2020).

Ice conditions found on site at the start of the drift experiment were exceptional to begin with: Record temperatures in summer and strong offshore-directed ice drift in winter resulted in the second longest ice-free summer period since reliable instrumental records began. As a result, ice thickness was unusually thin compared to the previous 26 years (Krumpen et al., 2020). However, by the time the MOSAiC floe reached Fram Strait (around 300 days later), it had grown to a thickness typical for the region in summer in the past decade (results from IceBird airborne surveys, Fig. 1 in Belter et al. 2021).

Satellite data played a decisive role for the campaign. Using a combination of satellite images acquired prior to the start, Krumpen et al. (2020) were able to follow the ice floe back to its place of origin, the shallow shelf of the New Siberian Islands (Fig. 1). During the drift itself, satellite images were continuously taken over the ship and the extended surroundings to support scientific objectives and logistic needs. Especially during the polar nights, these data became the only systematic source of information about the ice conditions in the wider area. In this manuscript we make use of satellite data records collected along the drift track to categorize the different ice conditions and most prominent events that shaped and characterised the floe and surroundings from October 4, 2019 to July 31, 2020. A comparison with previous years is made whenever possible for the reference period 2005/2006 – 2018/2019. The reference period was chosen such that it includes as many data products as possible. The aim of this analysis is to provide a very first and general overview of on-site conditions for upcoming physical, biogeochemical and ecological MOSAiC studies.

Below, we introduce the different satellite products used to describe the sea-ice conditions along the drift. A short description of the large-scale atmospheric pattern and its impact on the Transpolar Drift is provided. A more detailed description of the atmospheric conditions is given in Dethloff et al. (2021) and Rinke et al. (2021). Hereafter, we analyse the drift itself and reconstruct the course that the ship would have taken in previous years. This is followed by an analysis of ice concentration, ice thickness, snow thickness, and deformation and lead openings along the drift. Finally, we take a first glimpse at the distribution of melt ponds on the MOSAiC floe using high-resolution Sentinel-2 data collected before the CO entered Fram Strait and started to disintegrate. At the end of each chapter, key research questions are identified that should be addressed in future studies. In closing, we summarize the main findings.

## 2 Material and Methods

The Transpolar Drift carried *Polarstern* with the MOSAiC CO from its initial position on October 4, 2019 (85°N, 136°E) to Fram Strait (July 31, 2020, 78.9°N, 2°E) within 303 days. Hereafter, the ship was relocated to a position near the North Pole (87.7°N, 104°E on August 21, 2020), an area with limited satellite coverage. In this manuscript we therefore exclusively focus on the first phase of the MOSAiC expedition.

The 303 days long drift is reconstructed using GPS data from different sensors: From October 4, 2019 to May 15, 2020, we use the ship's GPS. Because *Polarstern* had to leave the CO temporarily from mid of May until June 18 for the purpose of crew exchange, this period was bridged with GPS data provided by a surface buoy deployed on the CO prior to departure

(buoy ID P225, [meereisportal.de,](meereisportal.de) last access: June 22, 2021). For the remaining period until July 31, we again use ship GPS data. In the following, the data products utilised to describe the ice conditions in the vicinity of the MOSAiC floe are introduced. An overview of the different products and their spatial and temporal resolution is given in Table 1. Where possible, we compare data in the full resolution of the respective satellite with mean values formed over a 50 and 100 km radius (Fig. 1, buffer).


## 2.1 Lagrangian sea-ice tracking

To investigate whether the 2019/2020 drift was comparable to previous years, we made use of satellite sea-ice motion data to reconstruct the pathways the ship would have taken if the experiment had started in one of the previous 14 years (October 2005 – 2018) instead. The satellite-based sea-ice pathways were determined with a drift analysis system called IceTrack. The

system traces sea-ice forward in time using a combination of satellite-derived, low-resolution drift products (Krumpen et al., 2019 and 2020, Belter et al., 2021). In summary, IceTrack uses a combination of three different ice drift products for the tracking of sea-ice: i) motion estimates based on a combination of scatterometer and radiometer data provided by the Center for Satellite Exploitation and Research (CERSAT, Girard-Ardhuin and Ezraty, 2012, 62.5 x 62.5 km grid spacing), ii) the OSI-405-c motion product from the Ocean and Sea Ice Satellite Application Facility (OSISAF, Lavergne, 2016, 62.5 x 62.5 km

grid spacing), and iii) Polar Pathfinder Daily Motion Vectors (v.4) from the National Snow and Ice Data Center (NSIDC, Tschudi et al., 2020, 25 x 25 km grid spacing). The IceTrack algorithm first checks for the availability of CERSAT motion data, since CERSAT provides the most consistent time series of motion vectors starting from 1991 to present and has shown reliable performance (Rozman et al., 2011; Krumpen et al., 2013). During summer months (June–July) when drift estimates from CERSAT are missing, motion information is bridged with the OSISAF product (2012 to present). Prior 2012, or if no

valid OSISAF motion vector is available within the search range, NSIDC data are applied. The reconstruction of "virtual" floes for these 14 years works as follows: Sea-ice at the starting position of the CO is traced forward in time on a daily basis starting on October 4 (1996 – 2019) until July 31 (303 days). Tracking is discontinued if sea-ice concentration at a specific location along the trajectory drops below 50%, which the algorithm defines as the position where the ice melted. The applied sea ice concentration product is provided by CERSAT (Ezraty et al. 2007) on a 12.5 km grid and is based on 85 GHz SSM/I

brightness temperatures, using the ARTIST Sea Ice (ASI) algorithm (Kaleschke et al. 2001).

To assess the accuracy of this Lagrangian tracking approach, Krumpen et al. (2019) reconstructed the pathways of 56 GPS buoys deployed between 2011 and 2016 in the central Arctic Ocean. The displacement between real and virtual tracks is approximately $36 \pm 20$ km after 200 days and considered to be in an acceptable range. To assess the accuracy of IceTrack in 2019/2020, we reconstruct the drift of the CO and 23 additional DN buoys (Krumpen and Sokolov, 2020). A comparison of

Fig. 2a with 2b shows that the reconstructed drift of the CO and other buoys is in close agreement with the observed drift. However, when the CO entered Fram Strait (red box), the reconstructed track lags behind the real one. The study of Krumpen et al. (2019) indicates that the limited performance of IceTrack in Fram Strait is likely the result of a general underestimation of drift speeds by low resolution satellite products in this area. It becomes particularly evident when looking at the reconstructed drift of the additional 23 DN buoys deployed in the vicinity of the CO (Fig. 2c). Within the first 200 days, the

reconstructed DN trajectories deviate only slightly from observed tracks ($28 \pm 15$ km after 200 days), but once the DN reaches Fram Strait (south of 82.5°N, after 250 days), the distance between real and reconstructed pathways is exponentially increasing. The comparison of the CO drift with the drift of the previous 14 years is therefore limited to the first 250 days.

## 2.2 Sea-ice concentration

A time series of sea-ice concentration along the MOSAiC trajectory between October to July (daily resolution) is obtained from the 89 GHz channels of the AMSR-E and AMSR2 microwave radiometer on the NASA Aqua and Jaxa GCOM-W satellites, respectively (Tab. 1, Spreen et al., 2008; Melsheimer & Spreen, 2019a,b). Data are available from [meereisportal.de](meereisportal.de)

and seaice.uni-bremen.de (last access: February 15, 2021). The spatial resolution of the dataset is 6.25 km grid spacing (the footprint sizes of the two sensors are with 4 and 5 km even smaller). The conditions in larger surroundings are determined by averaging all grid points falling within a 50 and 100 km radius (compare Tab. 1). For comparison with previous years, we extracted sea-ice concentration along the MOSAiC trajectory for the years 2005/2006 to 2019/2020. The year 2011/2012 is left out due to a gap between AMSR-E and AMSR2. Uncertainties (accuracy and precision) are usually below 5% for individual grid cells in winter and in the high ice concentration regime. In summer and at low ice concentration, uncertainties can be significantly larger (up to 25%; Spreen et al., 2008). Also, atmospheric influences like cloud liquid water and water vapor can affect the sea-ice concentration retrieved from 89 GHz channels. However, these are uncertainties of individual grid cells and mean biases for the averaged 50 and 100 km radii are lower if they are not affected by larger scale phenomena or summer melt. In summer or during warm air intrusions sea-ice concentration underestimation due to wettened ice surfaces, ice lenses or higher liquid water content in the snow, or melt ponds might occur. Such a period is observed during MOSAiC from mid-April to May 2020 and discussed below. During that time period, we show for comparison sea-ice concentration from an inverse multi-parameter retrieval based on AMSR2 data using optimal estimation (Scarlat et al., 2017, 2020). It uses all frequency channels from 7 to 89 GHz to retrieve seven surface and atmospheric parameters (including cloud liquid water and water vapor), which potentially can mitigate some of the effects like atmospheric influence causing wrong sea-ice concentrations for traditional single parameter satellite retrievals like the one introduced above. The spatial resolution of this dataset is approximately 40 km. During and following the warm air intrusion and the associated drizzle-on-snow event it shows more correct ice concentrations but is yet not available for the previous years and thus cannot be used as primary dataset here. Based on the 89 GHz sea-ice concentration dataset we also calculate the closest distance from the MOSAiC CO to the ice edge. To remove small openings in the ice we first smooth the sea-ice concentration dataset by convolution with a 4×4 (25 km) grid cell kernel, then the distances from the CO grid cell to all grid cells with zero sea-ice concentration are calculated and the shortest distance is selected as distance to the ice edge.

## 2.3 Sea-ice thickness

Sea-ice thickness (SIT) along the MOSAiC drift track during the Arctic winter season from October 2019 through April 2020 is analysed using two satellite remote sensing datasets. The first dataset is based on radar altimeter data from the CryoSat-2 (CS2) mission of the European Space Agency (ESA). We use SIT retrievals generated at the full resolution of the altimeter with an approximate point spacing of 300 m and swath width of 1650 m along the ground-track of the satellite (Table 1). The method of the SIT retrieval for each radar waveform is based on Ricker et al. (2014) with updates described in Hendricks et al. (2020). The dataset is named the AWI CryoSat-2 sea-ice thickness product version 2.3 and it is accessible through the website meereisportal.de (last access: February 15, 2021). In this study, we use the Level-2 pre-processed (l2p) product between October 1, 2019 and April 30, 2020. This processing level contains data from CryoSat-2 radar echoes along the ground tracks of all orbits within one day and SIT information is provided for each radar footprint of approximately 300 m times 1600 m in along and across-track direction respectively. Spatial averaging is necessary to reduce the significant retrieval noise for the individual radar echoes, but is not applied to the l2p data. Instead, subsets of all orbit data points within a day are generated based on their distance to the noon (UTC) position of the CO. For each subset we compute the mean SIT, the interquartile (IQR) and interdecile (ICR) SIT range, as well as the number of data points in each daily subset. According to the study logic, the search radius for the SIT subsets is chosen as 50 and 100 km and we only use individual orbits that provide at least 50 data points within the specified search radii. Both the number of l2p data points per day as well as their minimum distance to the *Polarstern* noon position are variable. The number of CS2 l2p data points for the 50 km (100 km) search radii varies from approximately 50 (300) at lower latitude to approximately 900 (2000) close to the maximum orbit coverage of CS2 at 88°N. No data within a short period in February 2020 is found at the 50 km search radius when the centre position of the search radius was above 88°N, while the 100 km search radius is sufficiently large to match CS2 orbits. We do not show

data from a smaller (e.g. 5 km) search radius, as very few orbits were close enough to the CO. For the same reason we also refrain from comparing short segments of l2p data to local observations on the CO, not only because of the lower temporal coverage but also because the retrieval noise in the l2p SIT data will dominate on the scale of the local SIT observations.

The second dataset used for the SIT estimation is the merged CryoSat-2/SMOS (CS2SMOS, version 203) SIT product (Ricker et al., 2017). CS2SMOS provides gridded SIT data at a resolution of 25 km, which is significantly lower than the CS2 l2p data, however the underlying optimal interpolation provides gapless SIT information, also north of the CS2 orbit limit of 88°N. CS2SMOS SIT estimates at the CO position during the short period when *Polarstern* drifted north of 88°N are thus based on a spatial extension of SIT gradients measured at the CS2 orbit limit. Each daily updated CS2SMOS SIT field is based on an observation period of 7 days and we use the centre of this period as the reference time to subset SIT data around the CO position at the selected radii. CS2SMOS data are based on CS2 l2p and Soil Moisture and Ocean Salinity (SMOS) SIT data. The SMOS retrieval provides thickness information of thin sea-ice, which complements the CS2 l2p data. The data merging uses a background field extending two weeks before and after the observation period, thus the temporal coverage is shorter than that of the CS2 l2p data and ranges from October 18, 2019 to April 12, 2020. In addition, the selection of SIT observations in the CS2SMOS data may vary from the CS2 l2p regional coverage as we use the grid cell centre positions within 50 km and 100 km radius around the CO to compute the daily mean CS2SMOS SIT value. The number of selected CS2SMOS SIT observations depends on the position of the CO relative to local grid cell coordinates. The number varies between 10 and 14 grid cells for the 50 km and between 47 and 52 for the 100 km search radius. We do not expect this variability to cause a selection bias due to the smoothness of the CS2SMOS SIT data.

## 2.4 Snow depth

Low resolution snow depth along the MOSAiC trajectory is retrieved from the 7 and 19 GHz channels of the AMSR-E and AMSR2 microwave radiometer following the method from Rostosky et al. (2018). Data is available via Rostosky et al. (2019a, b). Following Rostosky et al. (2020), uncertainties are based on Monte-Carlo simulations using varying input parameters for a snow and sea-ice (MEMLS, Tonboe et al., 2006) and atmosphere (PAMTRA, Mech et al. 2020) microwave emission model. Most sea-ice, snow, and atmosphere properties are not known to the satellite snow depth retrieval (only information about the ice type, multi-year or first-year, is provided based on Ye et al. (2016a,b)). Thus, by varying these properties and evaluating the influence on the snow depth retrieval, an estimate of the uncertainty caused by their unknown state can be obtained. The uncertainty range for snow depth on multi-year ice is between 5 to 10 cm on average (for individual grid cells it can be larger). The mean uncertainty estimate specifically for the MOSAiC dataset is 8 cm. The grid size of the snow depth data is 25 km. The snow depth retrieval for multi-year ice areas is currently limited to March and April, while for first-year ice it can be retrieved all winter (see Rostosky et al., 2018). As the MOSAiC ice floe was in an area of predominantly second-year ice, which radiometrically is considered multi-year ice for the snow depth retrieval, also snow depth for MOSAiC is only available for March and April. Here we present snow depth data for the grid cell centred at the CO (12.5 km radius) in addition to radii of 50 km and 100 km averages (compare Tab. 1). A comparison with previous years is made with snow depth data extracted along the MOSAiC drift path from 2005 until 2019.

## 2.5 Lead detection based on optical data

Sea-ice leads, i.e. lead frequencies and lead fractions along the MOSAiC drift track are derived from Moderate-Resolution Imaging Spectroradiometer (MODIS) thermal infrared data and the Collection 6 of ice surface temperatures (Hall and Riggs, 2019). In order to detect whether a lead is present in a certain pixel we employ the local surface temperature anomaly, which is expected to exhibit significant positive deviations when a lead is present during winter (November to April). This general procedure is followed by the application of a fuzzy inference system that assigns individual retrieval uncertainties to each detected lead pixel. Using this approach, we obtain daily categorical lead maps with separate classes for clouds, sea-ice, leads,

and artefacts, with the latter comprising detected leads with an uncertainty exceeding 30%. The full approach and the resulting products are described in Reiser et al. (2020). From this dataset we use daily lead data with a spatial resolution of 1 km for the months of November to April for the years of 2005/2006 to 2018/2019 (as the reference period) and for the winter of 2019/2020 (for MOSAiC, compare Tab. 1). The daily lead data can only be derived for winter months as the retrieval relies on a significant surface temperature contrast between leads and sea-ice. The derived lead frequency is a temporally integrated quantity that indicates how often a lead is found at a certain position within a defined period while the lead fraction is a spatially integrated quantity that provides the fraction of an area that was covered by leads. Note that days with a cloud fraction above 50% are excluded from the analysis.

### 2.6 Sea-ice deformation from high resolution radar images

In this study, we quantify sea-ice deformation based on sequential Synthetic Aperture Radar (SAR) scenes obtained by ESA's Sentinel-1A/B satellites along the drift track of the CO. Deformation is the consequence of divergence (opening), convergence (closing) and shear (sliding alongside) between ice floes. Regularly gridded sea-ice drift and deformation fields with a spatial resolution of 1.4 km are retrieved following the method described in von Albedyll et al. (2021). More details about the drift algorithm are provided in Thomas et al. (2008 and 2011) and Hollands and Dierking (2011). As input for the applied algorithm, we use HH-polarised scenes with a spatial resolution of 50 m. Images over the CO were taken during the entire MOSAiC drift, except for the period between January 14 and March 15, 2020, when the ship was north of the satellite coverage. The temporal resolution is typically one image per day (with few exceptions). Spatial derivates are calculated from the gridded velocity field and used to derive divergence, convergence, and shear (see von Albedyll et al. (2021) for details). To quantify deformation in the vicinity of the CO, we average all grid cells located within a 5 km radius around the ship. Exceptionally strong deformation events are defined as events with a magnitude exceeding two standard deviations of the 5 km average time series. To compare deformation in the vicinity of the ship with deformation over a larger area (50 km), averages are computed for 61 5 km circles arranged within a radius of 50 km around the ship (see illustration in Fig. 3). In this way we avoid biases due to scaling effects.

### 2.7 Characterization of melt pond coverage using optical Sentinel-2 data

To provide a first quantification of the spatial distribution and temporal development of large melt ponds on the MOSAiC floe, we downloaded all available Sentinel-2 (S2, ESA) satellite images (https://scihub.copernicus.eu/dhus/, last access: February 10, 2021) taken over the ship between the end of May and July 31, 2020.  Prior the end of May, the sun elevation was not high enough for passive optical remote sensing. A total of eight completely or partially cloud-free scenes could be identified. For the detection of melt ponds, we selected five scenes that are temporally equally spaced, namely June 21, July 1, July 7, July 22, and July 27, 2020. Next, the MOSAiC floe was clipped and a pond index was calculated by means of a normalised spectral index (e.g. Gignac et al., 2017, Watson et al., 2018) using S2 bands 4 (665 nm) and 8 (842 nm) as input. The pond index is used to differentiate between water and ice/snow. Note that only ponds larger than the spatial resolution of the S2 sensor (10 m) can be detected. We therefore assume that the actual pond cover is significantly underestimated, and that the method is only suitable for providing estimates of the timing and relative changes in pond coverage.

### 2.8 Reanalysis and ship weather data

Mean sea-level pressure, 2 m air temperature, and 10 m wind speed data for the time period 2005-2020 are taken from the newest version of the European Centre for Medium-range Weather Forecasts (ECMWF) global reanalysis, ERA5 (Hersbach et al. 2020). Hourly values along the MOSAiC trajectory in 2019/20 as well as in the preceding 14 years along the same trajectory are extracted by linear interpolation in time and space after triangulation of the rectangular 0.25° ERA5-grid. The 2019/2020 trajectory data are evaluated against corresponding standard meteorological observations on board of *Polarstern* (https://www.awi.de/nc/en/science/long-term-observations/atmosphere/polarstern.html, last access: February 22, 2021). The

ship measurements are however taken at non-standard heights (wind: 39 m; air temperature: 29 m; pressure: 16 m, reduced to sea level) so the evaluation is rather qualitative. More stringent comparisons of MOSAiC in-situ meteorological observations, not just from the ship but from a large number of sensors across the CO and DN, are beyond the scope of this paper but will

be conducted elsewhere.

## 3 Results and Discussion

### 3.1 Atmospheric conditions and the Transpolar Drift in 2019/2020

Large-scale surface air pressure and associated anomalies in 10 m wind speed (shown in Fig. 4) determined the course of the

MOSAiC drift and its deviation from the long-term average. October, November, and December were characterised by moderate monthly mean circulation anomalies, oriented mostly such that the winds (and thus the drift) were westward rather than northward, thereby preventing the MOSAiC floe from reaching the North Pole (note that Fig. 4 shows wind anomalies, whereas the drift path is the actual drift). Starting in January, large-scale low-pressure entered the Arctic from the European sector, resulting in an intensification of the Transpolar Drift in January, February, and March (Fig. 5), with the low-pressure

region gradually moving towards the Beaufort Sea. Correspondingly, these months were associated with an exceptionally high positive Arctic Oscillation (AO) index (see Dethloff et al. 2021 and Rinke et al. 2021 for a more detailed description). In April, the decaying low-pressure centre was located over the Beaufort Sea, resulting in a drift of the MOSAiC floe towards the Barents Sea. Next, the reversed air pressure gradient in May, with a high-pressure anomaly over the Beaufort Sea, pushed the MOSAiC floe towards northeast Greenland until it entered the Fram Strait area (June/July).

Figure 6 compares ERA5-based atmospheric conditions along the MOSAiC drift trajectory with conditions in the preceding 14 years. The circulation anomalies from January through May (Fig. 4) led to positive air temperature anomalies in northern Siberia and in the Kara and Laptev Sea, in particular in February (up to +10 K, not shown). In contrast, air temperature anomalies at the MOSAiC floe were rather moderate most of the time (Fig. 6 middle). Moderate warmer-than-average periods occurred in mid-November, late February, mid-April, and late May, whereas colder-than-average periods occurred in early

November and early March, with absolute minima around -35°C. Wintertime cold (warm) anomalies were typically associated with high (low) surface air pressure anomalies (Fig. 6 bottom). The positive AO months January, February, March, and April were accompanied by low pressure anomalies also at the MOSAiC floe (Fig. 6 bottom). High wind speeds were encountered in particular in these months but also in late November/early December (Fig. 6 top). Apart from these exceptions, meteorological conditions at the MOSAiC floe can be considered average compared to previous years.

The ERA5 data along the MOSAiC trajectory in 2019/2020 agree well with co-located ship observations (Fig. 7), in particular regarding surface air pressure. Wind speed tends to be slightly lower in ERA5, although it should be noted that the comparison with the raw on-board observations (e.g., winds are measured at 39 m instead of 10 m) has limitations. However, the winter warm bias in ERA5 over Arctic sea-ice of the order of 2-3 K (Fig. 7) is consistent with previous assessments (e.g., Batrak and Müller 2019). Note that the true 2 m air temperature bias might be even larger because the ship air temperatures might be

overestimated due to (i) local heat sources and (ii) higher temperatures at the measurement height of 29 m compared to 2 m in typical cases of near-surface inversion. Given that these differences are likely systematic and thus similar in other years, the anomalies discussed above are likely not strongly affected.

### 3.2 The MOSAiC drift and a comparison to previous years

We compared the drift of the MOSAiC floe with the course the CO would have taken if the experiment had started in any of the previous 14 years (October 2005 – 2018). The underlying satellite-based Lagrangian tracking approach is introduced in Section 2.1. Figure 8 summarises the results of this analysis. Panel a) shows the reproduced MOSAiC trajectory (multi-colored line) together with trajectories from previous years (grey lines). The large differences between the tracks show how difficult it is to accurately predict the course of a drifting platform and how large the spread of possible endpoints can be. Panel b)

provides the averaged satellite-derived daily displacement rates of the MOSAiC CO during the first 250 days as compared to the previous 14 years. With 8.52 km/d the drift speed in 2019/2020 is around 20% higher than the mean over the period from 2005 to 2018 (7.14 km/d ± 0.75). Only 2008/2009 shows an even higher average displacement rate (8.79 km/d), although Fram Strait is reached a few days later due to a more northerly route. Another striking year is 2018/2019, with only average daily displacement rates, but a strong westward drift component, which would have carried the ship even faster toward Fram Strait than in 2019/2020. A trend towards a faster Transpolar Drift as reported by Spreen et al. 2011 or Krumpen et al. 2019 cannot

be deduced from this rather simple and spatially limited analysis. However, results shown here are in line with these studies.

### 3.3 Sea-ice concentration

Sea-ice concentration along the MOSAiC drift trajectory in 2019/2020 and the reference period 2005/2006 to 2018/2019 is

shown in Figure 9. The average sea-ice concentration between October 4, 2019 to July 31, 2020 amounts to 97% (based on the 89 GHz sea-ice concentration (black line; Spreen et al., 2008), i.e., containing the underestimation in April discussed below). The seasonal evolution is characterised by a substantial temporal variability over the course of the 303-day long drift. This variability is almost independent of the spatial scale used with only minor differences (±0.5% deviation from mean) between the sea-ice concentration values determined from the 3, 50, and 100 km radius (Fig. 10).

Given the high agreement between the values from different radii, we focus in the following discussion on the time series with the highest resolution (3 km radius, Fig. 9). The October to July sea-ice concentration average along the MOSAiC drift trajectory agrees well with the long-term 2005/2006 to 2019/2020 average (both have a mean of 97%). However, on shorter time scales there are significant differences: During the first half of the drift (October until end of February) the MOSAiC ice concentration was with 99.5% about 1% higher than the long-term average (compare black with blue line, Fig. 9), while during

the second half (March until end of July), it was lower than during the long-term average and shows higher variability than the first half. High ice concentration like the 99.5% are not unusual (compare to the grey lines) and can be expected in winter in the Central Arctic (e.g., Kwok, 2002). The second half with lower (actually false) ice concentration is more unusual and will be discussed further in the following.

Sea-ice concentration variability stayed below 5% until March 2020, when first significant reductions in ice concentration

occurred. At this time, the CO was already positioned north of Fram Strait and the distance to the ice edge was gradually decreasing (compare Fig. 11). With the onset of spring in March/April first major drops in ice concentration below 90% occurred. The strong ice concentration reductions down to 75% from mid-April until mid-May (average ice concentration 87%) were due to a false satellite ice concentration retrieval. Visual observations from the ship's bridge confirm that the ice concentration on average stayed higher than 95% during that time period. We can see that at that time a warm air intrusion

raised temperatures close to 0°C, which was accompanied by a significant increase in wind speed (Fig. 6). The warming induced strong temperature gradients and increased vapor fluxes in the snow, which can cause stronger snow metamorphism and significantly change the snow permittivity already at above −5°C snow temperatures (Mätzler, 1992). Also, liquid water content can increase at temperatures slightly below 0°C and small liquid water fractions of, e.g., 2% strongly change the microwave loss in the snow (Hallikainen, 1986). Refreezing after the warming event can cause ice lenses in the snow. Such

events were previously observed to an influence on microwave properties and penetration (e.g., King et al., 2018). On April 2019 slight drizzle was observed, which likely refroze on the snow afterwards. These surface processes and additional weather influence by high water vapour and cloud liquid water affect the microwave polarization difference (e.g., Lu et al., 2018) and likely caused the strong fluctuation in ice concentration for the ASI algorithm used here. Other ice concentration algorithms for AMSR2 satellite data (e.g. NASA-Team) showed similar effects (not shown). As an alternative we present the ice

concentration from an optimal estimation retrieval (Scarlat et al., 2018, 2020) during that critical time period in Figure 9, which attempts to take such effects into account (specifically the atmospheric influence) and in our case is in better agreement with the ship-based observations. Also, in previous years (grey lines, Fig. 9) occasionally ice concentrations below 90% were

observed in the sea-ice concentration record during mid-winter. We have not investigated if these were real openings in the ice caused by ice divergence or atmosphere induced effects like in our 2020 case. After mid-May 2020 the ice concentration recovered to almost 100%. In July, the floe started to disintegrate and ice concentration dropped to 85% within a radius of 3 km around *Polarstern*, and below 60% in the 50 and 100 km radii (Fig. 10).

We determine the closest distance to the ice edge from sea-ice concentration maps (Fig. 11). At the beginning of the MOSAiC expedition, the distance from the CO to the ice edge was about 320 km. During October the distance gradually increased to 1000 km due to the freeze-up of the Russian marginal seas. Once the MOSAiC CO approached Fram Strait (March 2020) the distance to the ice edge steadily decreased until the ice margin was reached at the end of July 2020. Note that the winter variability in ice edge distance was caused by polynya activity in the Russian shelf seas.

Future studies will investigate the impact of the warm air intrusion on microwave properties in more detail based on the extensive in-situ microwave and snow/ice measurements conducted on the MOSAiC floe. However, also the smaller fluctuations of the sea-ice concentration between 97% and 100% during October to February need further investigation by combining them more closely with the different lead fraction and ice divergence records discussed below. This will help to investigate the partitioning between thermodynamic and dynamic redistribution of ice mass as well as the impact of ocean to atmosphere heat fluxes.

### 3.4 Sea-ice thickness

Both satellite-based sea-ice thickness products show the expected increase in ice thickness between October 2019 and April 2020 (Fig. 12 and Table 2). Except for the period between February 14 to March 8, 2020, when the CO was positioned north of 88°N, the high orbit density of CS2 allows almost continuous daily coverage at 50 and 100 km radius. The monthly mean thickness within a 50 km (100 km) radius around the CO changed from 0.77 m (0.8 m) in October 2019 to 2.40 m (2.51 m) in April, 2020. The sea-ice thickness distribution is characterised by the IQR (difference between 75% and 25% percentile) and the IDR (difference between 90% and 10% percentile, compare methods). The increase in sea-ice thickness was accompanied by a similarly increased IQR and IDR, indicating a wider sea-ice thickness distribution as a result of thermodynamic ice growth and deformation of the older ice class and the formation of young ice throughout the winter season. Specific dynamic events sensed by other remote sensing sensors have a visible impact on the change of mean SIT and thus the apparent SIT growth rates. Lead formation in mid-November 2019, also seen in a strong divergence event, which added new thin ice coincides with an intermittent SIT decrease (Figure 19). The SIT distribution in the second half of April 2020 also widened significantly at a time when both lead fractions and a drop in sea-ice concentration indicates the presence of new ice formation.

It is also notable that the CS2 L2P sea-ice thickness was, on average, consistently thinner at the 50 km radius compared to the 100 km radius (Table 2, on average 6 cm (4%) thinner between October and April). Similarly, IQR and IDR were larger for 100 km than for 50 km, however the larger number of data points in the wider search area may also lead to a higher likelihood of diverse sea-ice conditions. This is in agreement with findings of Krumpen et al. (2020). According to the authors, the MOSAiC DN was set up at a regional thickness minimum. The local minimum is related to the ice age: Sea-ice in the DN was formed three weeks later than the surrounding ice. However, Krumpen et al. (2020) report even larger differences in sea-ice thickness of 36% between the DN area and areas further away.

Results from CS2SMOS mirror these findings of thinner ice close to the CO compared to the larger scale, though differences are smaller (Fig. 12, Table 2). This can be expected, as the primary input to the CS2SMOS analysis in the central Arctic is CS2 data due to its higher sensitivity to thicker ice than SMOS. The main differences to CS2 l2p are therefore the influence of SMOS in the beginning of the winter and the larger degree of smoothing introduced by the optimal interpolation. The monthly mean sea-ice thickness values in Table 2 are therefore mainly consistent with the exception of October and November 2019. In this period, CS2SMOS was consistently higher by approximately 0.15 m with respect to the CS2 l2p data.

We do not expect that the locally lower thicknesses in the DN are well represented in the CS2SMOS SIT, since these are influenced by a larger region due to the interpolation method. The CS2 l2p thicknesses instead are effectively point measurements at kilometre scale and are apparently able to pick up the local thickness gradient with thickness differences smaller than the uncertainty of absolute SIT values. The discrepancy between the CS2 l2p and CS2SMOS thicknesses persisted well into November 2019 and became less prominent afterwards. This provides evidence that the local thickness minimum at the MOSAiC DN became less prominent over the winter season, though still at a detectable level as indicated by the consistently but minor differences at radii of 50 and 100 km.

Since CS2 l2p and CS2SMOS are in general consistent over the winter season, we use CS2SMOS data to compare sea-ice conditions during the MOSAiC drift with the past nine winter seasons in the CS2SMOS data record (Fig. 13). The comparison between the years shows a comparably low sea-ice thickness in the 10-year long data record at the location of the MOSAiC expedition, if not the lowest for segments in the earlier part of the drift. The monthly sea-ice thickness during MOSAiC was approximately 0.4 m lower at the beginning of the drift compared to mean monthly CS2SMOS of all previous winters (Table 2). The differences reduced towards 0.3 m in April, indicating slightly stronger thermodynamic and dynamic ice growth with respect to the average, potentially aided by the thinner sea-ice in the beginning. These results are however based on a SIT data record that depends on climatological values for snow load and sea-ice density and thus does not contain the impact by the expected variability of these parameters in the SIT retrieval. For example, using dynamic snow load in SIT retrieval by satellite radar altimeter has resulted in a more pronounced interannual variability but also stronger thickness trends in the Arctic marginal seas (Mallett et al., 2021). While MOSAiC has taken place in the central Arctic with generally thicker ice and snow, a similar impact can be expected as well. We therefore consider it unlikely that the differences between the SIT estimates along the MOSAiC drift tracks for the 10 years of CS2SMOS data can be explained by retrieval uncertainty alone. Field observations with longer time series are needed to evaluate the stability of SIT retrievals over decadal periods (e.g., Khvorostovsky et al., 2020), which are not available for the location of MOSAiC.

CS2SMOS also indicates sea-ice thickness differences between the 50 km radius and the 100 km radius showing that the MOSAiC expedition took place in a local sea-ice thickness minimum. It should be noted that the SIT differences, specifically between the two search radii were well below the uncertainty estimate of the retrieval for both CS2SMOS and CS2. Gridded CS2 data indicate a retrieval uncertainty on average of 0.5 and 0.7 m between October 2019 and April 2020 at a scale of 25 km and monthly periods (Hendricks and Ricker, 2020). The main driver of the uncertainty magnitude however is less retrieval noise but the uncertainty of auxiliary parameters such as snow load and sea-ice density. The deviation between actual values of these parameters and their parametrizations in the satellite retrieval is likely to have larger correlation length scales. Thus, the satellite sensors might be able to sense local SIT differences, though the absolute SIT uncertainty remains substantial. The finding of consistently thinner ice for the 50 km search radius compared to the 100 km search radius throughout the drift might be seen as a demonstration of this point.

Future work with the MOSAiC field data will focus on improving accuracy of SIT retrievals as well as quantifying its true magnitude. But given the sensitivity of present-day SIT products to local thickness differences and their sensitivity to dynamic events captured by other sensors, the question remains how SIT data at high spatial and temporal resolution can be used to better observe and understand the dynamics of the sea-ice cover.

### 3.5 Snow depth

Figure 14 shows a time series of satellite-based snow thickness in March/April for the years between 2005 and 2020. The mean March-April snow depth during the MOSAiC year was 22 cm at the 12.5 km radius (22/23 cm in 50/100 km radius) with an uncertainty of 5 cm. Note that the observed snow thickness during MOSAiC is around 3 cm lower than the long-term average of the period 2005 to 2019. A preliminary comparison (not shown) of satellite-based snow thickness estimates with

in-situ observations from the MOSAiC CO indicates a good agreement with errors not exceeding the expected uncertainty of on average 5 cm.

The snow depth during MOSAiC was a few centimetres lower but overall, quite average compared to the long-term mean. The time series in Figure 14 shows that the snow depth stayed almost constant from beginning of March until mid-April. Only after the warm air intrusion in April (Fig. 6), increased precipitation led to a small increase in snow depth of about 3 cm. This is in agreement but potentially a bit lower than the detected snowfall by several sensors in the MOSAiC CO (about 10–20 mm snow water equivalent, i.e., approx. 4–8 cm snow depth; Wagner et al., 2021). However, the Wagner et al. (2021) study also shows that snowfall does not always directly relate to snow depth increases because lateral snow redistribution plays a significant role. Future studies will evaluate the satellite snow depth in more detail based on the extensive snow measurements taken during MOSAiC.

The satellite AMSR-E/2 March/April 2020 snow depth of 22 cm is significantly lower than the snow climatology from Warren et al. (1999) for the years 1954 to 1991. For that the March/April snow depth for the MOSAiC region would have been between 35 and 39 cm, i.e., 60% to 80% higher than during MOSAiC and the whole AMSR-E/2 time period 2005 to 2019 (green line in Fig. 14). Thus, we observe a strong reduction in snow depth for the MOSAiC region compared to previous decades. This also has implications for ice thickness retrievals from satellite altimeters, where the Warren snow depth climatology often is used for the freeboard to ice thickness conversion (Section 2.3 and e.g., Ricker et al., 2014).

Here we only present one satellite-based snow depth product. Future studies will compare our snow depth retrievals from the AMSR-E/2 microwave radiometers with snow depth from combined CryoSat-2 and ICESat-2 measurements (Kwok et al., 2020) and snow depth from SMOS (Maaß et al., 2013).

## 3.6 Leads

The mean winter lead frequency (November to April between 2005/2006 and 2018/2019) for the central Arctic Basin and adjacent seas is shown in Figure 15a. The climatology shows that between November and April the central Arctic Ocean is generally characterised by low lead frequencies with values of roughly 0.1. This agrees well with consistently high ice concentration values indicated by the sea-ice concentration climatology during the first half of the expedition (Fig. 9). According to the climatology, higher lead frequencies (> 0.15) in winter are only to be expected near the ice edge and in Fram Strait. The lead frequency anomalies for the MOSAiC year 2019/2020 shown in Fig. 15b indicate no significant deviations from the winter mean climatology. On average, anomalies were slightly negative along the MOSAiC drift trajectory and in the sector between 30°W and 120°E, which again agrees well with the observed slightly higher ice concentration values as compared to the long-term mean (Fig. 9).

Regional differences in lead frequencies can be inferred from monthly lead anomaly maps shown in Fig. 15c-h. The monthly maps reveal anomalously high lead frequencies north of Greenland and Ellesmere Island between November 2019 and January 2020. Moreover, the strong positive anomalies in the Barents Sea in January 2020 and in the Beaufort Sea in February/March 2020 are worth mentioning (compare Dethloff et al. 2021). However, in the proximity of the CO no significant lead anomalies were found between November 2019 and February 2020. Only in March, when the CO was crossing a region of East-West oriented leads, slightly higher anomaly values of up to 0.1 are indicated (compare Fig. 4). In April 2020, leads around the MOSAiC CO were more North-South oriented and strengthened as expressed by higher anomaly values of up to 0.2 (Fig. 15h).

A detailed view on the temporal evolution of lead fractions along the MOSAiC drift trajectory on different radii is presented in Fig. 16. However, meaningful conclusions can only be drawn for the periods in which the cloud fraction for the respective radii was below 50%. Note that in Fig. 16 days with missing data and higher cloud fractions are indicated by red dots. Lead fraction is shown for the area around the MOSAiC CO with 10 km, 50 km, and 100 km radius, together with the mean and maximum lead fraction for the reference period and one standard deviation. The mean lead fraction for the area around the CO

was slightly increasing towards the end of winter for all of the three ranges shown, which confirms the drift into a region with generally higher average lead frequencies starting in March (Fig. 16a). In general, lead dynamics around the MOSAiC CO were typical for the respective region and point in time with only short, but significant deviations from the mean. A maximum in lead activity was observed on March 4 (at all radii). Several smaller events with lead fractions exceeding one standard deviation from the reference period were recorded on December 11 – 12, January 19, January 28, February 1, February 4 – 8,

March 1 – 5, March 11, and April 23 - 24 (for 50 km radius, Fig. 16a). Note that these events were only to some extent accompanied by a decrease in ice concentration, which might be explained by e.g. differences in spatial resolutions and different thin-ice sensitivities of the respective sensors.

### 3.7 Sea-ice deformation

Figure 17 shows the time series of divergence, convergence, and shear rates along the MOSAiC drift track at 5 km and 50 km radii as obtained from Sentinel-1 SAR data. Overall, we find that deformation close to the ship (5 km radius), was representative for the deformation experienced by the ice cover at larger distances (up to 50 km). Despite the different geographical regions we find that mean shear and combined divergence and convergence of 8% $d^{-1}$ and 2% $d^{-1}$ along the MOSAiC drift track are in good agreement with deformation rates obtained from a ship-radar north of Svalbard during the N-

ICE2015 drift campaign (Oikkonen et al., 2017).

The variability of divergence, convergence, and shear showed a seasonal behavior which is linked to the consolidation of the ice pack and in agreement with findings of previous studies (e.g., Itkin et al., 2017, Hutchings et al., 2011.). Monthly averages of the time series indicate that deformation was moderate and balanced in convergence and divergence in the consolidation phase between October and November 2019 (Fig. 17). Hereafter, divergence, convergence and shear temporarily decreased

from December 2019 to January 2020. In March to May 2020, divergence, convergence, and shear went back to a moderate level until a sudden increase in June and July was observed when the MOSAiC CO approached the marginal ice zone. Note that monthly averaged divergence correlates reasonably well with intensified lead activity observed by optical satellites (Sect. 3.6). In spring (March, April), the ice experienced more divergent than convergent motion, which again agrees well with intensified lead activity observed in spring (Fig. 15/16).

On daily time scales, divergence, convergence, and shear were characterised by long quiet phases occasionally interrupted by strong deformation events (video supplement). The average temporal spacing between such deformation events was 2.5 weeks. However, the events were not uniformly distributed in time, as 60% of the events took place between October and November (grey bars in Fig. 17). The strongest deformation event within the 50 km radius of the CO was observed on April 14-17, 2020. By that time, a lead of almost 2.5 km width opened up at 25 km distance of the CO (Fig. 3).

We expect that future MOSAiC studies will investigate the driving processes behind seasonal and short-term deformation events in more detail, using a combination of on-ice, airborne and ship-based observations (e.g. stress measurements, airborne laser and thickness surveys and ship radar sequences) and data from various satellite products. We suggest the following future research questions: How does the ice thickness influence divergence, convergence, and shear in response to the wind forcing, and what is the role of convergence in creating a thick ice cover and how does deformation shape the ice thickness distribution?


### 3.8 Melt pond distribution

Figure 18 presents five cloud-free S2 scenes obtained between June 21 and July 27, 2020 that provide a first overview of the temporal and spatial evolution of melt ponds on the MOSAiC floe and its extended surroundings. Melt pond coverage is characterised using the pond index described in Section 2.7, where high values indicate water and low values ice/snow.

One of the most striking features is that at the time when the first cloud-free scene (July 21) was taken, large melt ponds had already developed on the MOSAiC floe. The earlier start of melt pond formation on the MOSAiC floe as compared to the extended surrounding is likely related to the surface topography. Compared to the surrounding floes, the MOSAiC floe was

characterised by heavily deformed areas which may have favoured early accumulation of large meltwater ponds. Another possible reason for the early onset of melting may have been the high quantity of sediments that were trapped in the ice (Krumpen et al. 2020). The high sediment content temporarily reduced the surface albedo of the floe, which may have favoured early melt of ice.

Within the following ten days, the proportion of large melt ponds on the MOSAiC floe increased considerably, and large ponds also began to form on the neighbouring floes. On July 7, while the total amount of melt ponds was still increasing, a few large melt ponds began to drain. In the final scene (July 27), taken almost three weeks after the draining began and just before the floe was abandoned, large melt ponds had mostly split into smaller ponds and had partially disappeared as a result of several drainage events that were observed in field between July 1 and 27. The (absolute) quantification of melt pond fraction is limited, as the typical size of the melt ponds observed on the ground were equal to or smaller than the pixel size of the S2 image.

The unusual temporal and spatial evolution of melt ponds on the MOSAiC floe compared to the surrounding floes raises the question of what processes preconditioned the early melt. More specifically, what role did the heavily deformed area play in the formation of melt ponds and to what extent did the presence of sediments accelerate melting processes?

**4 Conclusion**

Below we summarize the ice conditions along the drift of the MOSAiC floe and the extended surroundings and compare them to previous years (2005/2006 – 2018/2019). The analysis is based on satellite data products commonly used for the scientific analysis of sea-ice in the Arctic Ocean. A summarizing overview of the atmospheric and sea-ice conditions observed along track is given in Fig. 19.

- A comparison of the MOSAiC trajectory with reconstructed satellite-based pathways for the past 14 years indicates that the drift during the first 250 days of the expedition was around 20% faster than the climatological mean drift. Deviations from a long-term average drift path are to a large extent the consequence of prevailing large-scale low-air-pressure anomalies, which resulted in an intensification of the Transpolar Drift between January and March 2020.

- CS2 and CS2SMOS data show that the mean thickness of sea-ice around the CO (50 km radius) evolved from 0.77 m in October 2019 to 2.40 m in April 2020. Sea-ice near the CO (50 km radius) was thereby 4% thinner as compared to surrounding sea-ice (100 km radius). According to Krumpen et al. (2020), the negative anomaly is due to the younger ice age, as the ice around the CO was formed in a different region and later in the year than the surrounding ice. A comparison with CS2SMOS records from the past nine winters shows that the ice around the MOSAiC CO was comparatively thin (partially the thinnest). In October 2019 it was 0.4 m and in April 0.3 m below the nine year average.

- Unlike ice thickness, snow thickness did not differ significantly from the long-term mean. Data from satellite-based microwave radiometers indicate an average March/April 2020 snow depth of 22 cm (12 km radius). This is 3 cm lower than the long-term mean for the years 2006 to 2019.

- From the start of the expedition until April, the average ice concentration within the 50 km radius of the CO was slightly higher (1%) than the long-term mean with low variability but occasional drops in ice concentration by up to 3%, which would impact ocean-atmosphere fluxes. In April and May a wrong reduction in sea-ice concentration is observed in some sea-ice concentration products, as a result of a positive air temperature anomaly, which changed the microwave properties of the snow but did not melt it. A significant drop in ice concentration took place at the end of the first expedition phase when the floe approached the ice edge (July).

- An analysis of winter (October – April) lead frequencies inferred from MODIS thermal infrared data indicates no significant deviation in lead activity from the mean climatology (2005/2006 – 2018/2019). At most, a slight negative deviation from the winter mean is discernible, which agrees well with the positive anomaly in ice concentration

between October and April. It is interesting to note that with increasing variability in ice concentration from March onwards, lead activity increased.

- A deformation time series derived from Sentinel-1 data gives first insights into divergence, convergence, and shear events along the MOSAiC drift path. Overall, we find that sea-ice deformation on the 5 km radius including the MOSAiC CO was representative for the wider (50 km radius) surroundings. Deformation rates were lower during winter, and higher during summer, which is in agreement with observations from previous studies. The dominance of divergence during spring agrees well with the observed higher lead fractions.

- Five cloud-free S2 scenes obtained during the melting phase provide insight into temporal and spatial evolution of melt pond coverage on the MOSAiC floe. Particularly worth mentioning is that formation of melt ponds began earlier on the MOSAiC floe than on neighbouring floes.

**Acknowledgement**

This work was supported by the German Ministry for Education and Research (BMBF) as part of the Russian-German Research Cooperation WTZ-RUS QUARCCS (grant 03F0777A) and CATS (grant 03F0831C), the International Multidisciplinary drifting Observatory for the Study of the Arctic Climate (grant MOSAiC20192020, AWI_PS122_00), IceSense (grants 03F0866B and 03F0866A), and the Seamless Sea Ice Prediction project (SSIP; grant 01LN1701A). We acknowledge support by the Deutsche Forschungsgemeinschaft (DFG) project 268020496–TRR 172, within the Transregional Collaborative Research Center "Arctic Amplification: Climate Relevant Atmospheric and Surface Processes, and Feedback Mechanisms (AC)[3]. ERA5 data were generated and provided by ECMWF and the Copernicus Climate Change Service. The work on satellite remote sensing data was partly funded through the EU H2020 project SPICES (640161), the ESA Sea Ice CCI phase 1 and 2 (AO/1-6772/11/I-AM) and the Helmholtz PACES II (Polar regions And Coasts in the changing Earth System) and FRAM (FRontiers in Arctic marine Monitoring) program. The production of the CryoSat-2/SMOS sea-ice thickness data was funded by the ESA project SMOS & CryoSat-2 Sea Ice Data Product Processing and Dissemination Service (ESA Contract No. 4000124731/lS/I-EF).

**Code/data availability**

Ship-based meteorological observations used in this manuscript were obtained as part of MOSAiC with the tag MOSAiC20192020 and the Project_ID AWI_PS122_00 and are available via PANGAEA. The gridded CryoSat/SMOS datasets are available via ftp://ftp.awi.de/sea_ice/product/cryosat2_smos/v203/nh/ and documentation can be found at https://earth.esa.int/eogateway/catalog/smos-cryosat-l4-sea-ice-thickness. The sea-ice concentration and snow depth data are available at https://seaice.uni-bremen.de and until 2018 from PANGAEA (Melsheimer & Spreen, 2019a, 2019b). This work contains modified Copernicus Sentinel data [2019/2020]. Sentinel-1 scenes are available from the Copernicus Open Access Hub (https://scihub.copernicus.eu/dhus/home).

**Competing interests**

The authors declare no competing interests.

**Author contributions**

T.K., L.v.A., H. G., S. H, B. J., G. S. & S. W. conceived the study and wrote the paper. J. B., C. H., L. K., C. K., X. T. K., R. R., J. R., S. S. & J. S. provided data, contributed to the analysis and discussion.

**Video Supplement**

Time series of divergence, convergence, and shear fields along the drift track of the MOSAiC floe from October 5, 2019 to July 14 2020. Ice drift is displayed as arrows while deformation is shown as colors. The title states the time period for which the deformation was calculated. The white circles around the *Polarstern* position have 5 and 50 km radii (https://doi.org/10.5446/51302)

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

**Table 1:** List of parameters and products investigated along the MOSAiC trajectory, with their respective satellite platforms and data distributors, as well as their resolution (temporal and spatial) and investigated radii and corresponding terminology.

| Parameter / Product | Satellite | Distributor | Investigated period | Product resolution | | Investigated radii (km) | | |
|---|---|---|---|---|---|---|---|---|
| | | | | Temporal (d) | Spatial (km) | close | medium | far |
| Sea-ice concentration | AMSR-E/AMSR2 | Uni Bremen | 2005/2006 - 2019/2020 | 1 | 6.25 | 3.125 | 50 | 100 |
| Sea-ice thickness, CS2 l2p | CryoSat-2 | AWI | 2011/2012 - 2019/2020 | 1 | 0.3 | 0.3 | 50* | 100* |
| Sea-ice thickness, CS2SMOS | CryoSat-2/SMOS | AWI | 2011/2012 - 2019/2020 | 7 | 25 | 12.5 | 50 | 100 |
| Snow thickness | AMSR-E/AMSR2 | Uni Bremen | 2005/2006 - 2019/2020 | 1 | 25 | 12.5 | 50 | 100 |
| Sea-ice lead frequency | MODIS | Uni Trier | 2005/2006 - 2019/2020 | 1 | 1 | 10 | 50 | 100 |
| Sea-ice deformation | Sentinel-1A/B | AWI | 2019/2020 | ~1 | 1.4 | 5* | 50* | - |
| Sea-ice melt pond coverage | Sentinel-2 | AWI | 2019/2020 | ~1 | 0.01 | - | - | - |
| Sea-ice trajectories derived from various motion products * | Multiple satellites* | OSI-SAF, NSDIC, CERSAT | 1995/1996 - 2019/2020 | 1 | 10 - 62.5 km | - | - | - |

* see method description for details

| | | | | Product resolution | | Investigated radii (km) | | |
|---|---|---|---|---|---|---|---|---|

**Table 2:** Monthly statistics of sea-ice thickness (SIT) from CryoSat-2 (CS2) level-2p (L2P) orbit and gridded CryoSat-2/SMOS (CS2SMOS) data for two radii around the CO position. The CS2 SIT distribution is characterised with the interquartile range (IQR) as difference between 75% and 25% percentile and the interdecile range (IDR) as difference between 90% and 10% percentile. For CS2SMOS, the SIT difference ($\Delta$SIT) between the MOSAiC year and SIT from the same drift trajectory but of previous winters since 2010 is given. The asterisk (*) marks that the mean SIT of CS2SMOS depends on less years than the other month since the CS2SMOS data record only starts in November 2010.

| | CS2 L2P | | | | | | CS2SMOS | | | |
|---|---|---|---|---|---|---|---|---|---|---|
| | SIT (m) | | SIT IQR (m) | | SIT IDR (m) | | SIT (m) | | $\Delta$SIT (m) | |
| | 50 km | 100 km | 50 km | 100 km | 50 km | 100 km | 50 km | 100 km | 50 km | 100 km |
| Oct 19* | 0.77 | 0.80 | 0.47 | 0.51 | 0.93 | 1.04 | 0.95 | 0.97 | -0.41 | -0.38 |
| Nov 19 | 1.02 | 1.07 | 0.52 | 0.60 | 1.05 | 1.23 | 1.13 | 1.15 | -0.45 | -0.43 |
| Dec 19 | 1.26 | 1.31 | 0.57 | 0.62 | 1.14 | 1.27 | 1.35 | 1.37 | -0.38 | -0.35 |
| Jan 20 | 1.46 | 1.48 | 0.61 | 0.63 | 1.21 | 1.28 | 1.50 | 1.51 | -0.38 | -0.36 |
| Feb 20 | 1.90 | 1.99 | 0.69 | 0.79 | 1.39 | 1.60 | 1.99 | 2.00 | -0.29 | -0.28 |
| Mar 20 | 2.23 | 2.27 | 0.85 | 0.88 | 1.74 | 1.81 | 2.31 | 2.33 | -0.43 | -0.41 |
| Apr 20 | 2.40 | 2.51 | 1.21 | 1.24 | 2.33 | 2.40 | 2.50 | 2.51 | -0.29 | -0.27 |


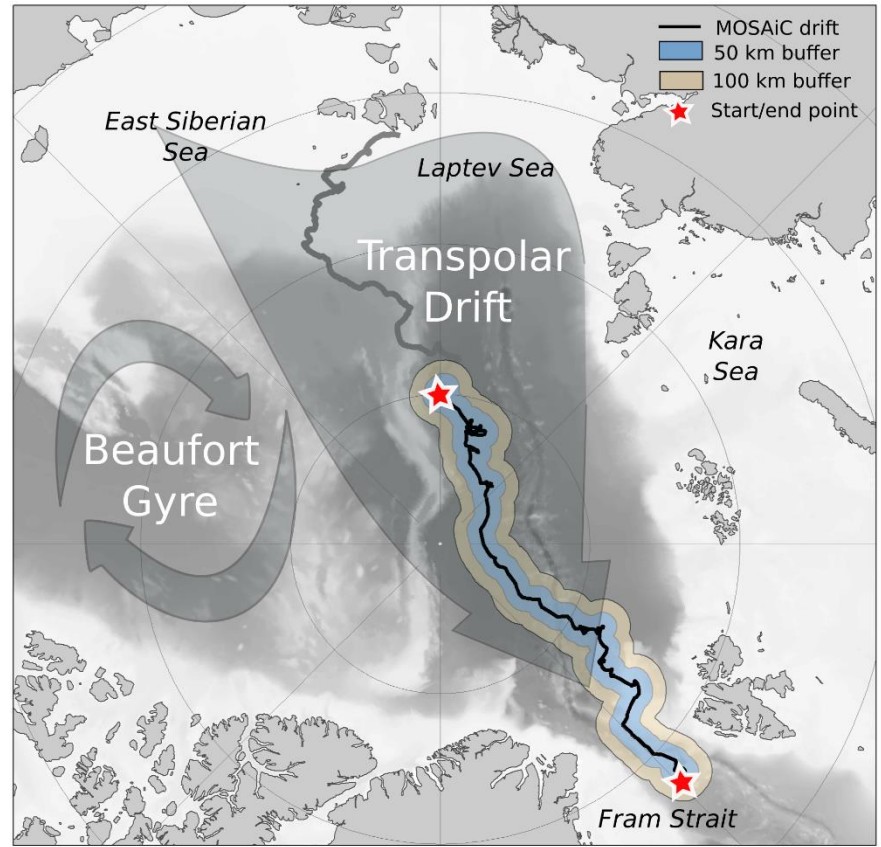

**Figure 1:** The MOSAiC drift (black line, the Central Observatory, CO) with a 50 km (blue) and 100 km (orange) buffer. Start (October 4, 2019) and end point (July 31, 2020) of the first MOSAiC expedition phase are indicated by red stars. Following Krumpen et al. (2020), the MOSAiC floe originated from the New Siberian Islands (grey thick line). The bathymetry of the Arctic Ocean (grey colors in the background) is based on Jakobsson et al. (2012)

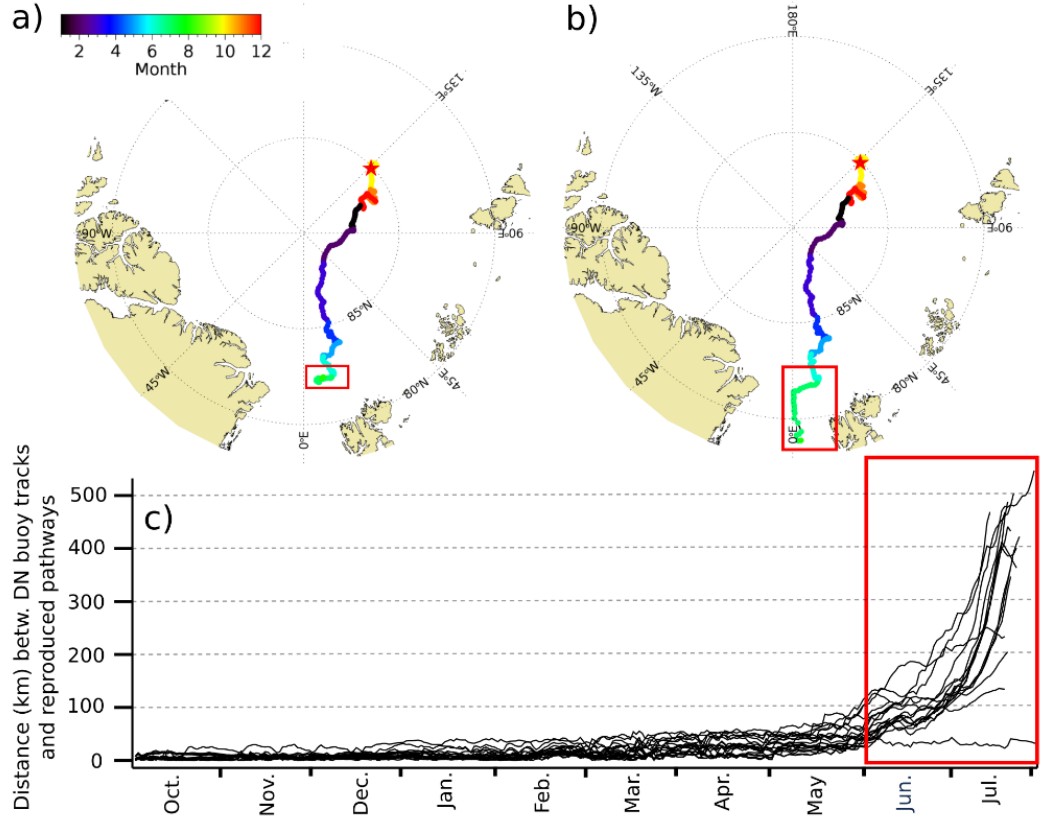


**Figure 2**: Comparison of MOSAiC CO and DN buoy tracks with IceTrack results: a) Reproduced pathway of the CO with IceTrack, b) Real (GPS-based) track of the CO, c) Distance between 23 DN buoys (source: seaiceportal.de) deployed on sea ice in the vicinity of the CO at the beginning of October 2019 and their reconstructed trajectories. Deviation between real and virtual tracks is small. Only once buoys enter Fram Strait (beginning of June (day 240) at 82.5°N), the distance is gradually
increasing (red box).

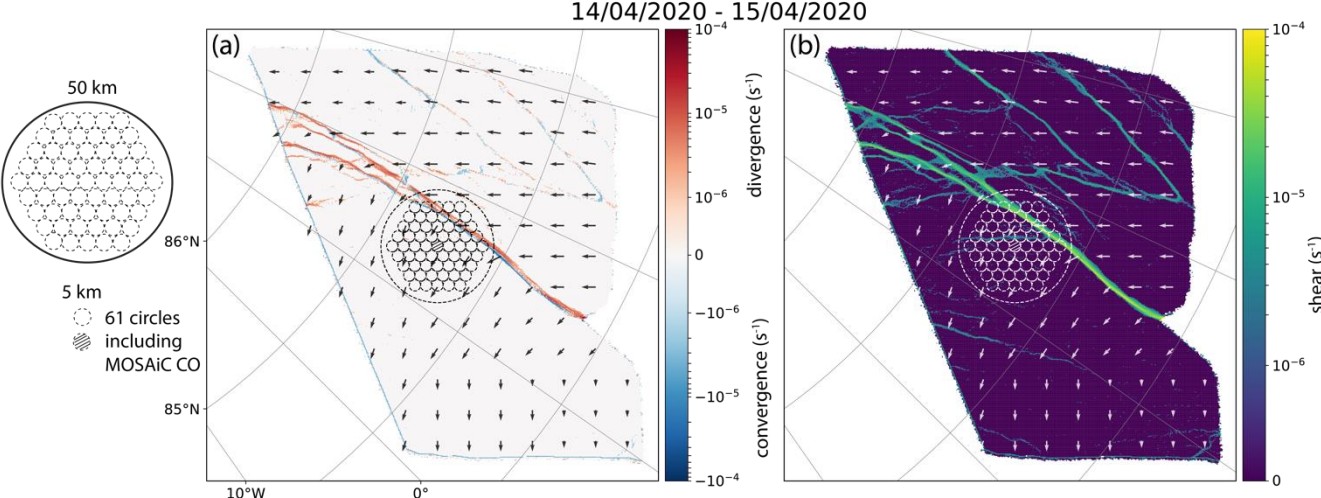

**Figure 3**: Left: To compare deformation in the vicinity of the ship (5 km) with deformation on larger scales (50 km), averages were computed for 61 5 km circles arranged within a radius of 50 km around the ship. Centre/right: Example of divergence, convergence (middle) and shear (right) derived from two consecutive Sentinel-1 SAR images acquired on April 14 (07:26:14) and 15 (08:07:03) 2020. Sea ice motion is displayed as black arrows. The image pair shows the strongest deformation event observed: Within 24 hours, a 2.5 km wide north-south oriented lead opened up ~25 km away from the CO.

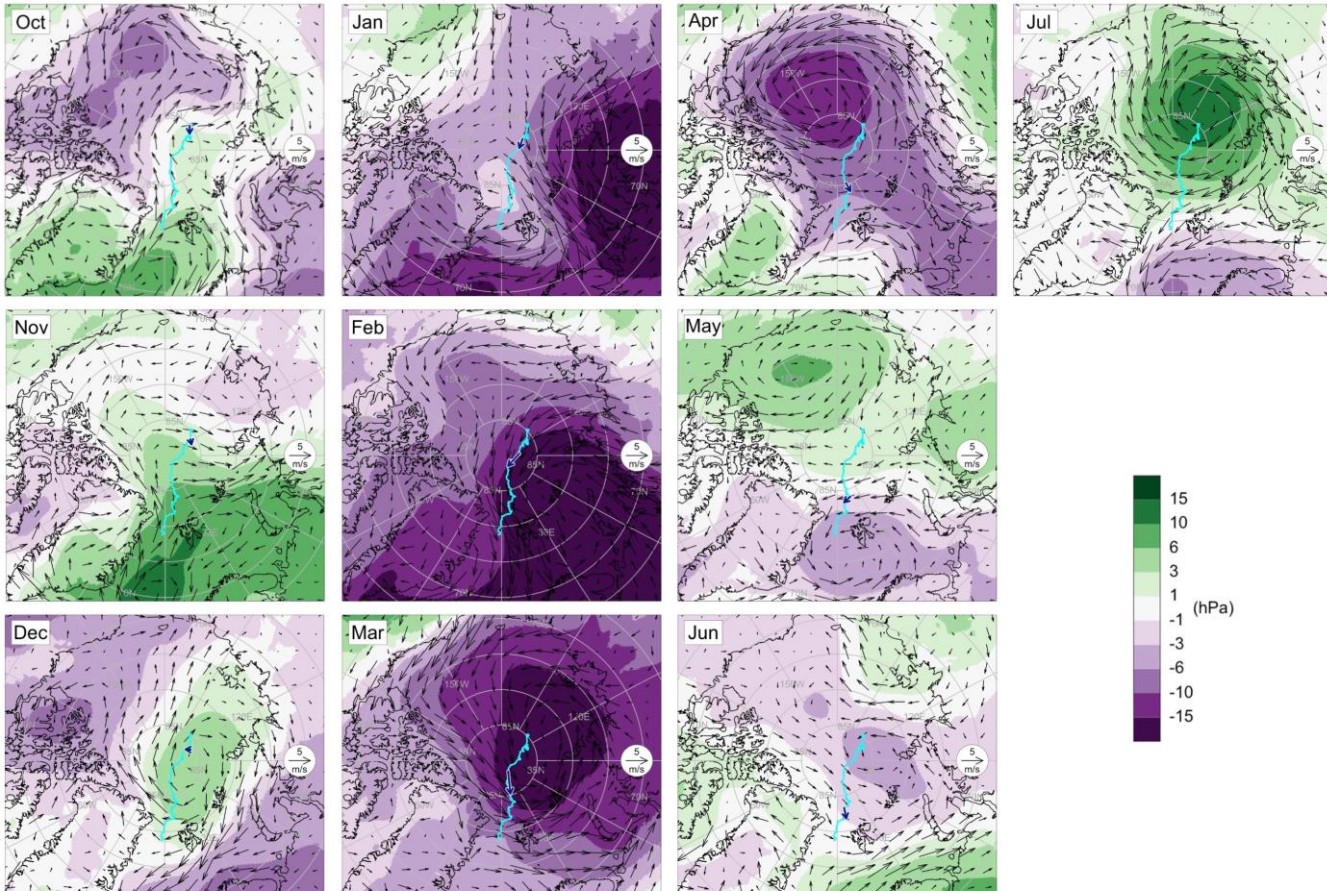

**Figure 4:** Monthly-mean sea-level air pressure (shading) and 10 m wind (arrows) anomalies with respect to the reference period 2005-2019 for each month of the MOSAiC drift from October 2019 to July 2020. The complete drift path is denoted by cyan lines; the drift during the respective month is denoted by blue arrows.


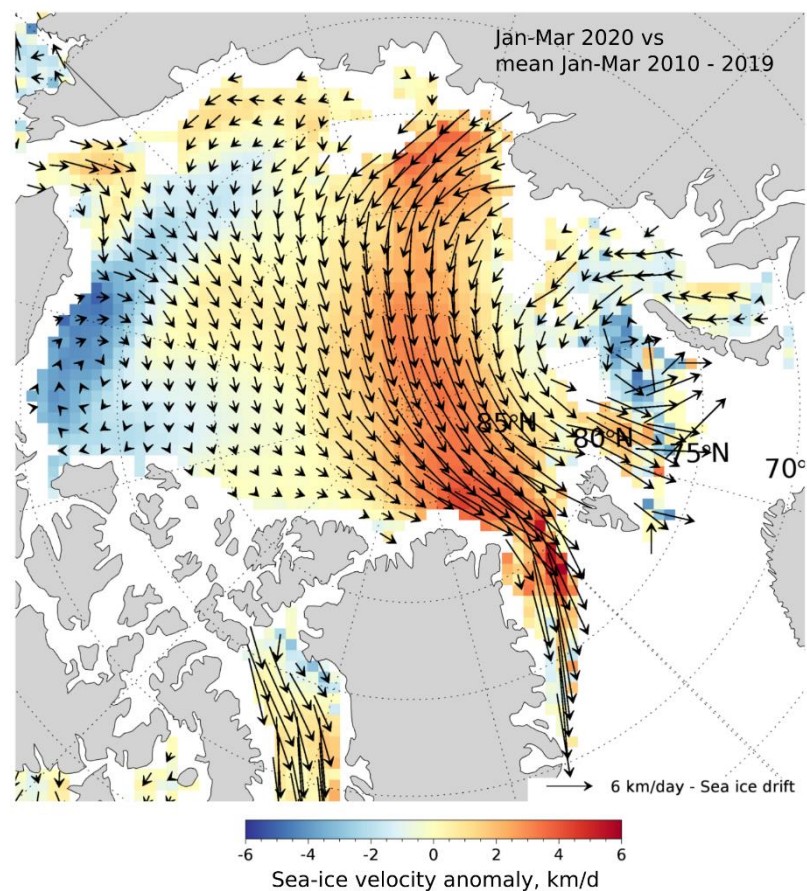

**Figure 5:** Three-month (January – March) sea-ice velocity anomalies in 2020 with respect to the reference period 2010-2019. Anomalies were computed from the OSI-405-c motion product provided by the Ocean and Sea Ice Satellite Application Facility (OSISAF, Lavergne, 2016). The vectors plotted on top indicate the average daily sea-ice motion for the same period (reprint from Dethloff et al. 2021).

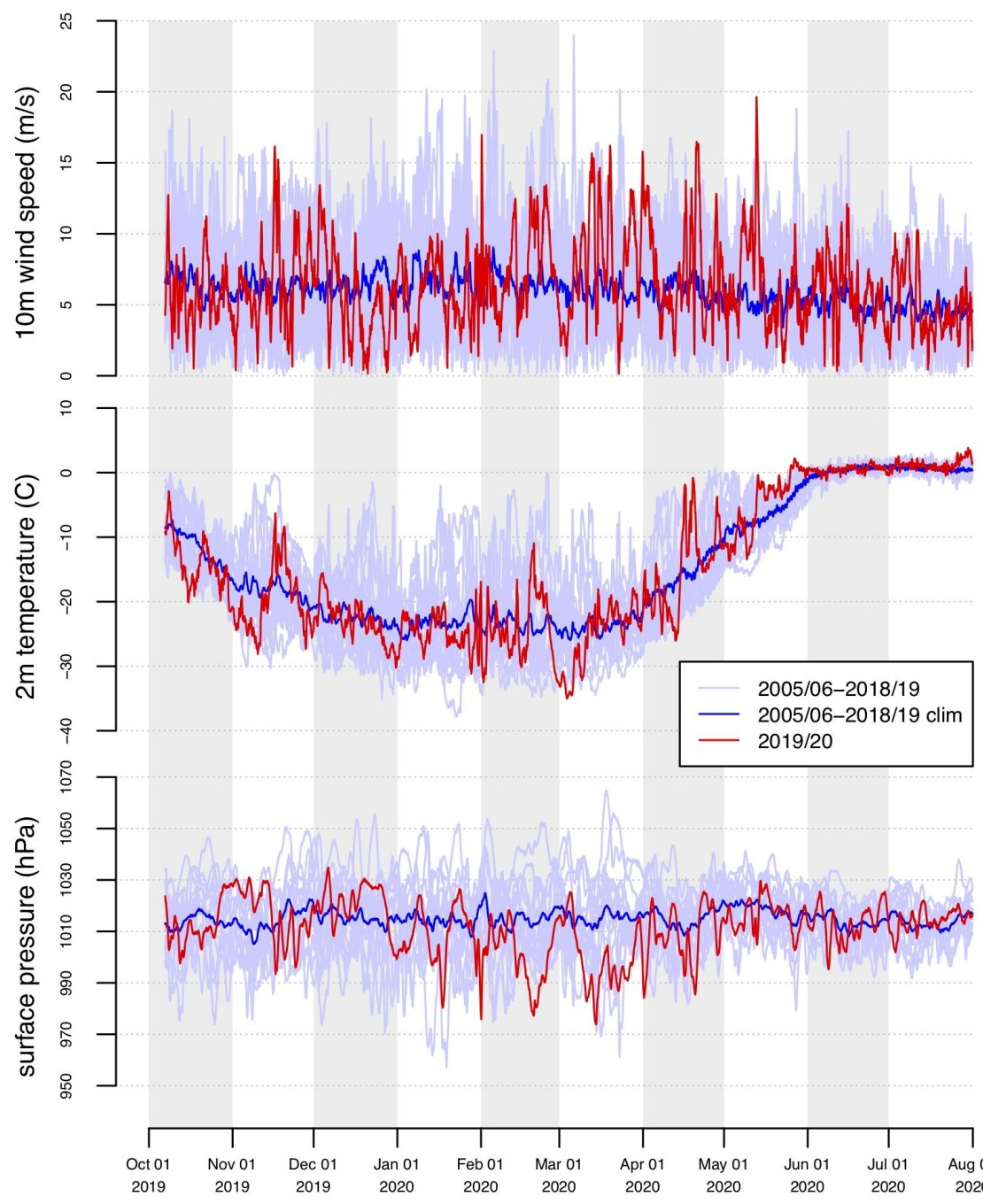


**Figure 6:** Hourly atmospheric conditions along the MOSAiC drift trajectory according to ERA5 in 2019/2020 (red) and in the preceding 14 years (light blue; average in dark blue). Top: 10 m wind speed; Middle: 2 m air temperature; Bottom: surface air pressure. See Fig. 7 for a comparison with corresponding ship observations.

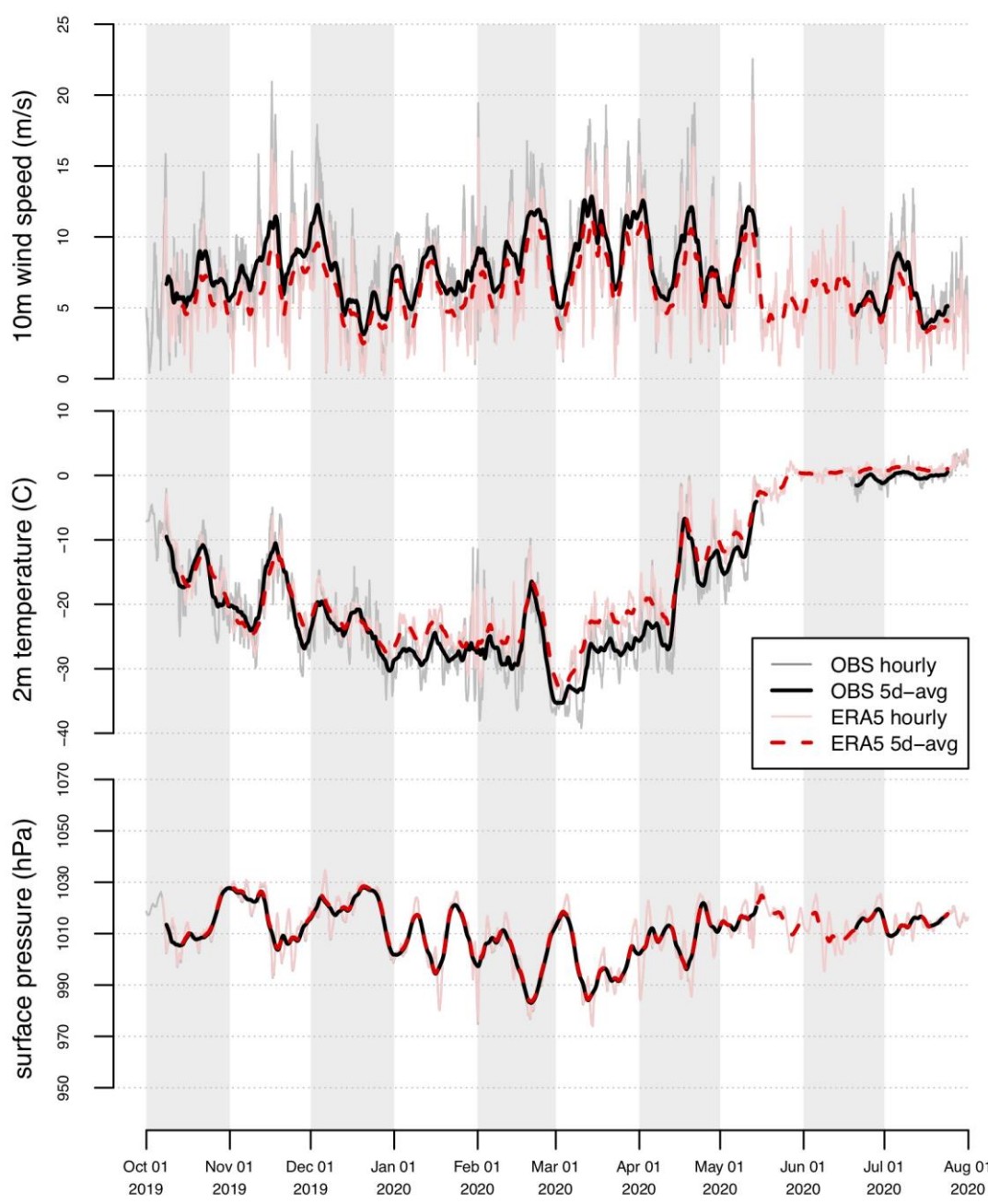


**Figure 7:** Atmospheric conditions along the MOSAiC drift trajectory in 2019/20 according to ERA5 (red and light red) and according to ship measurements (black and grey). Hourly data are depicted in light red and grey; 5-day-averages are depicted in red (dashed) and black. Top: 10 m wind speed; Middle: 2 m air temperature; Bottom: surface air pressure.


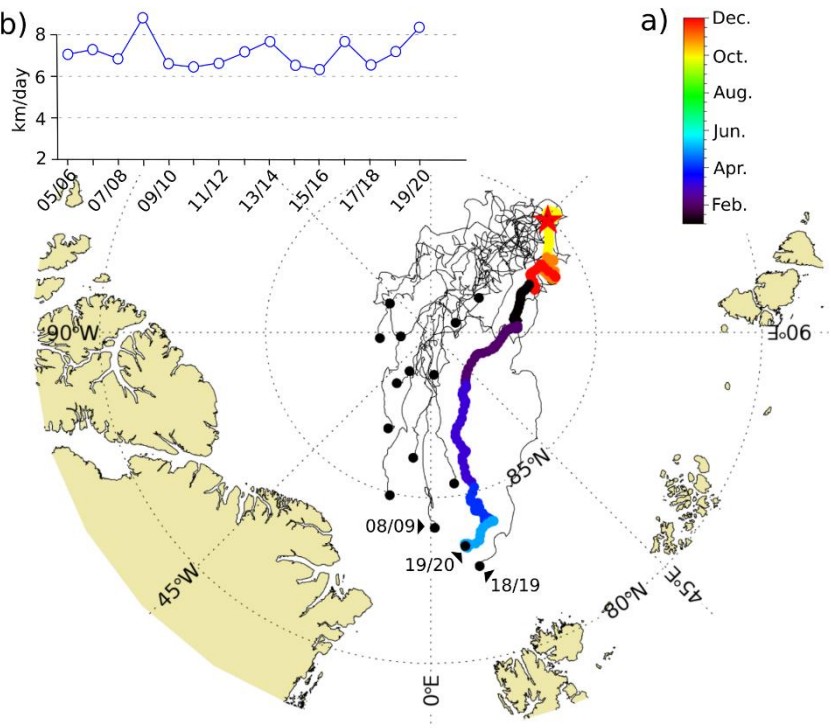

**Figure 8:** A comparison of the MOSAiC drift with the drift of previous years. a) Results from a forward tracking experiment: Sea ice was traced 14 times in a forward direction for a period of 250 days starting on October 4 (2005 – 2019) from the position where the *Polarstern* drift started (red star). The multicoloured trajectory line, with colours corresponding to the month of year, indicates the reproduced drift of the MOSAiC CO (Central Observatory). All other years (2005 – 2018) are shown as black lines. The end nodes of the individual tracks are marked by a black circle. b) Averaged displacement of sea-ice per day (km) for the individual years.

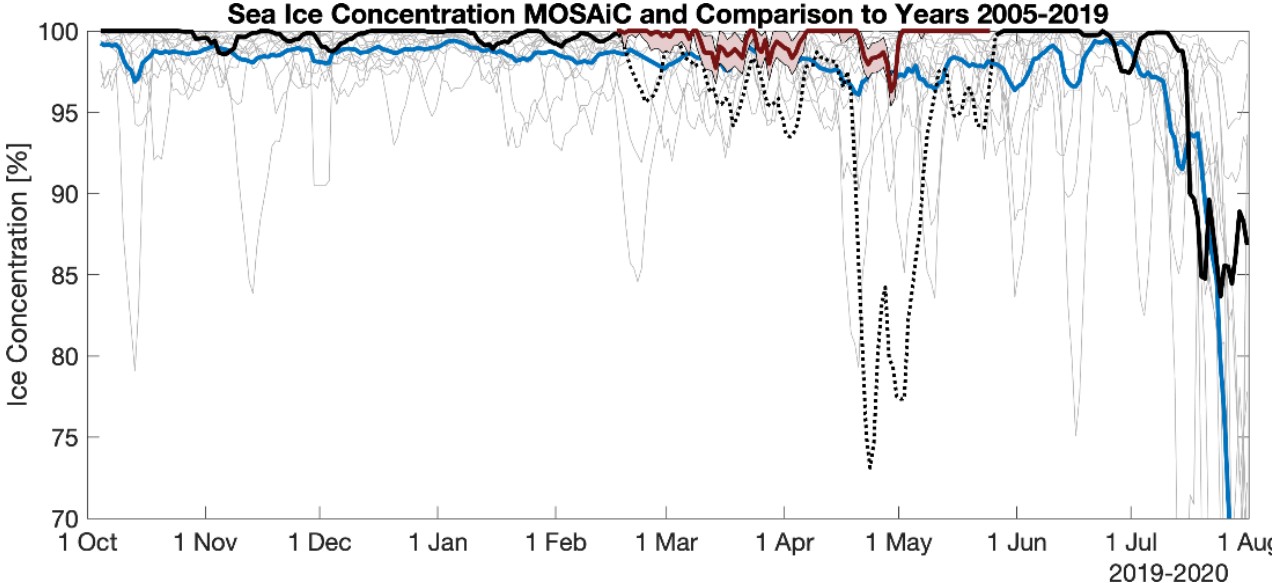

**Figure 9**: Sea-ice concentration within a 3 km radius around the CO (6.25 km grid cell, black) along the MOSAiC drift from October 4, 2019 to July 31, 2020 in comparison to the ice concentrations from 2005/2006 to 2018/2019 for the same drift trajectory. The blue line shows the average for 2005/2006–2018/2019 while the grey lines show the individual years. All timeseries are smoothed with a 5-day running mean. During spring warm air intrusions caused a significant temporary reduction of the sea-ice concentration (dashed black line). We therefore show in addition an alternative sea-ice concentration dataset during that time with uncertainty estimates (red and shaded red; not available for the climatology, see main text).


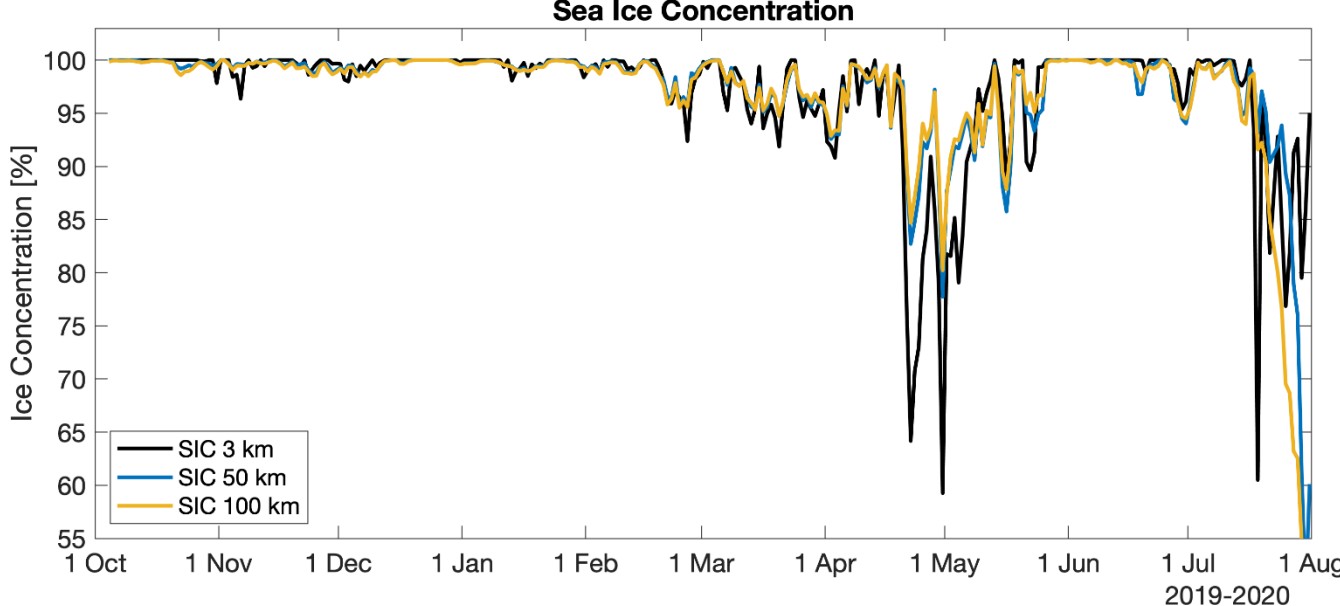


**Figure 10**: Sea-ice concentration along the MOSAiC drift trajectory from the start of the drift on October 4 2019 until the end of the first floe on July 31 2020. Daily (no smoothing) sea-ice concentrations are shown at 3.125 (black), 50 (blue), and 100 km (yellow) radius. Note the significantly underestimated concentrations between mid-April to May and associated discussion in the main text and Figure 9.


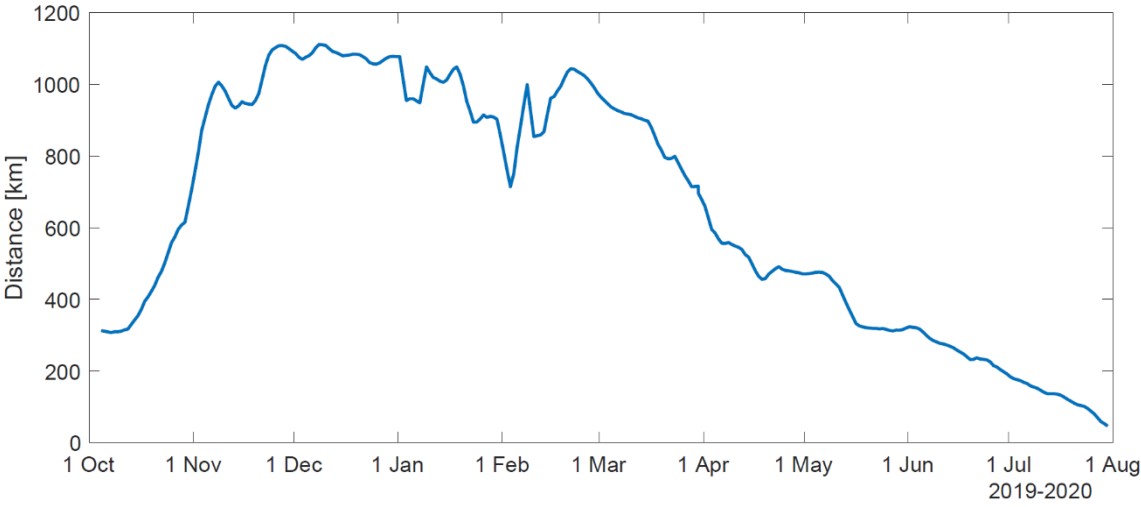

**Figure 11**: Distance of the MOSAiC CO to the ice edge obtained from the sea-ice concentration dataset.


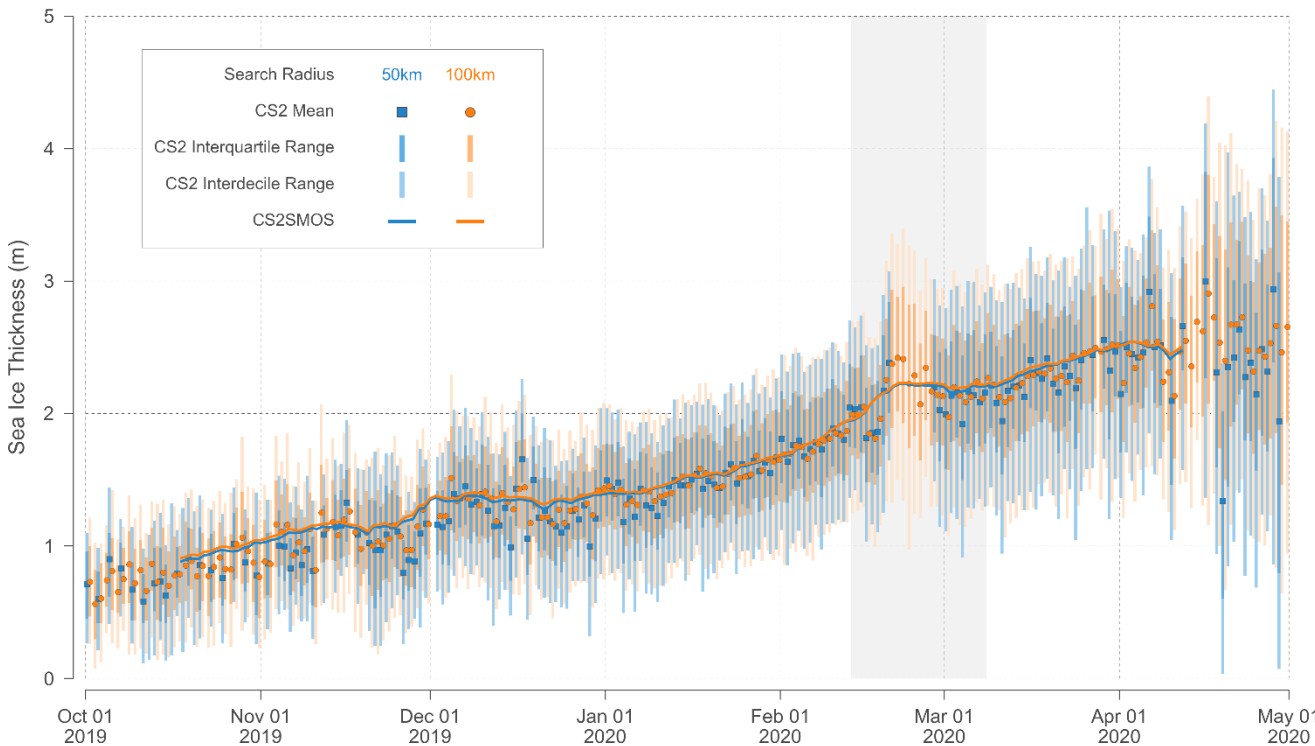

**Figure 12**: Daily sea-ice thickness estimates from CryoSat-2 (CS2) full resolution orbit L2P data and gridded CryoSat-2/SMOS (CS2SMOS) multi-sensor thickness analysis extracted for two different search radii (50 and 100 km) centered around the noon position of the CO for each day of the drift. Results from L2P data is only present for days where at least 50 L2P data points are found in both search radii. The grey rectangle indicates when the CO drifted north of 88°N and outside the CS2 orbit coverage. The distribution of CS2 orbit data within the search radii is described by the mean value, interquartile (25% to 75% percentiles) and interdecile ranges (10% to 90% percentiles). For CS2SMOS, only the mean values of grid values within the search radius are provided.

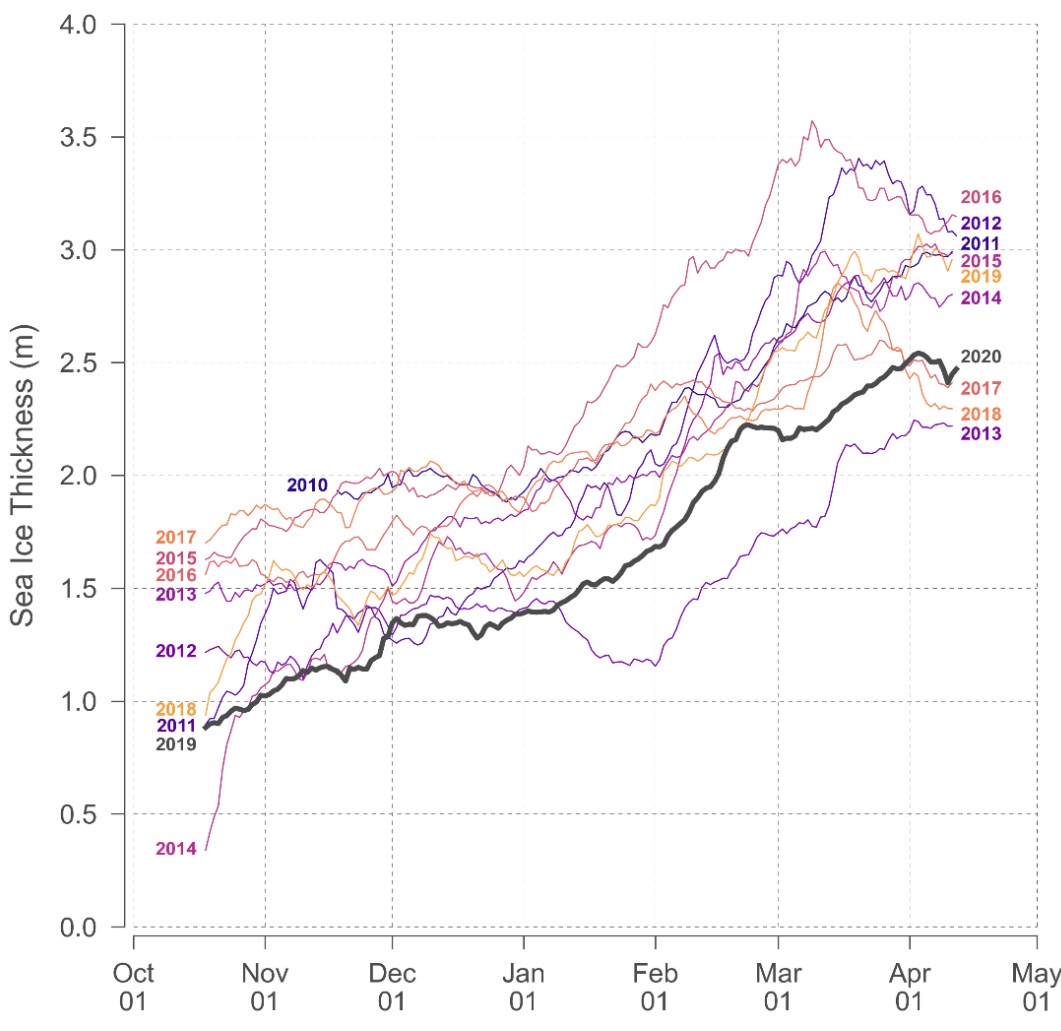


**Figure 13**: Daily sea-ice thickness from gridded CryoSat-2/SMOS (CS2SMOS) multi-sensor thickness analysis extracted within 50 km of the CO noon position for all Arctic winters in the CS2SMOS data record. Each winter season is marked by the start and end year, e.g. 2011/2012. Bold black line indicates data during the MOSAiC year and is identical to the 50 km radius CS2SMOS data in Fig. 12.


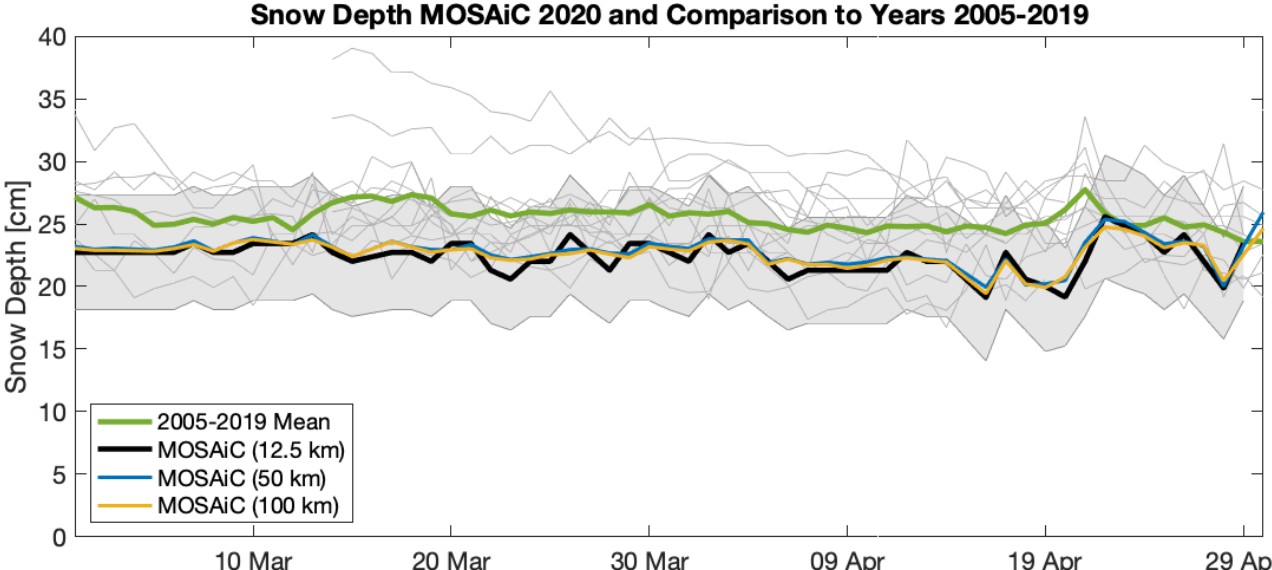

**Figure 14:** Snow depth from the AMSR-E and AMSR2 satellite microwave radiometers from March 1 to April 30 of the years 2005 to 2020 (without 2012) at the MOSAiC location. For the MOSAiC year 2020 the thick black line shows snow depth at 12.5 km radius. The grey shaded area indicates the corresponding uncertainty bounds (Rostosky et al, 2018, 2020). The blue and yellow line gives the mean snow depth at a 50 and 100 km radius for comparison. The thin grey lines show the individual years 2005 to 2019 and the green line provides the climatological mean. Because the MOSAiC floe was located in an area with partly second year ice, snow depth can only be retrieved for March and April (Rostosky et al, 2018).


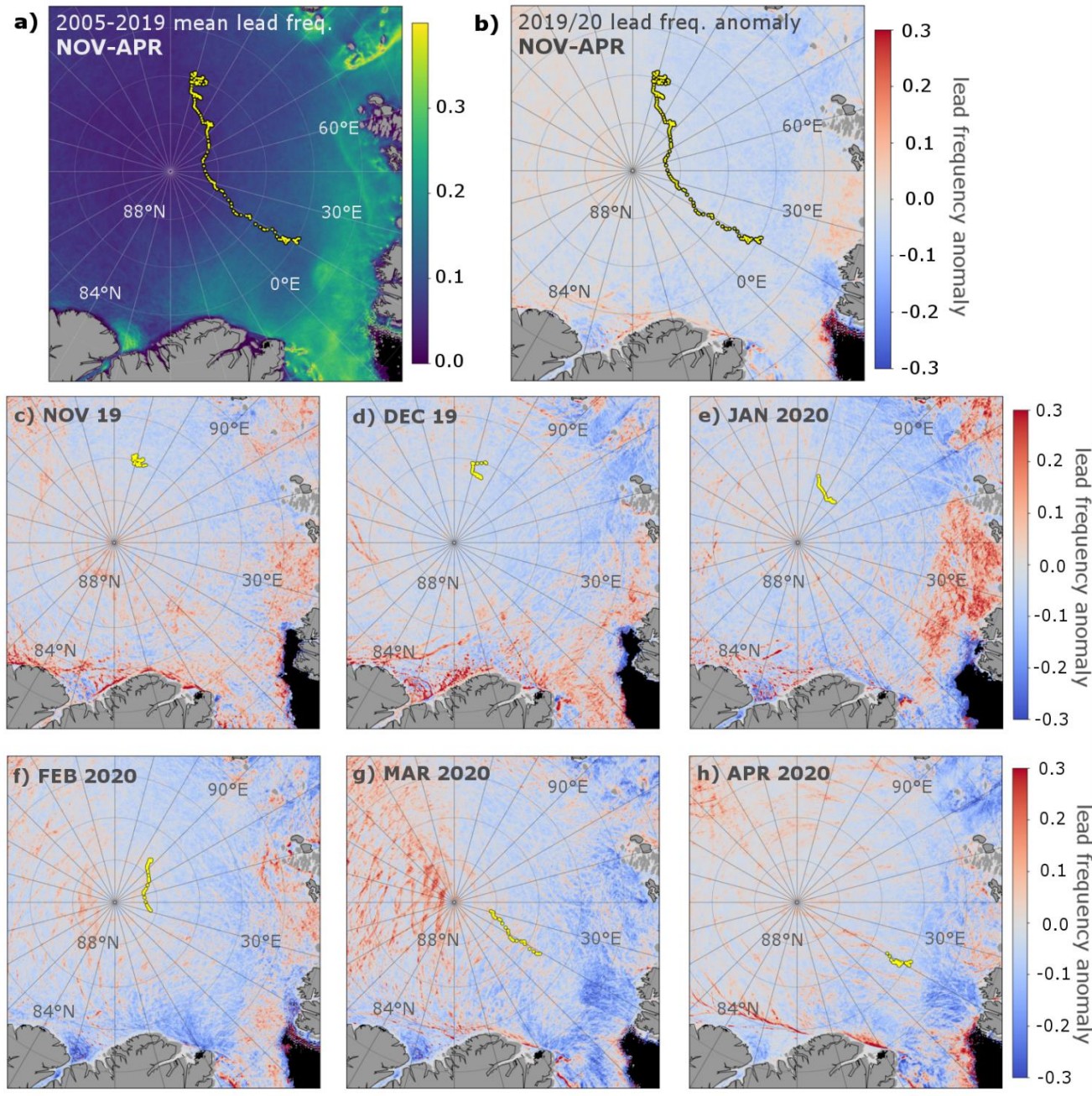

**Figure 15**: a) Spatial distribution of the mean frequency of occurrence of sea-ice leads for the months of November to April for the winter seasons 2005/2006 to 2018/2019 (reference period). MOSAiC drift of the CO is shown by yellow dots; b) lead frequency anomaly for the MOSAiC winter 2019/2020 with respect to the reference period, c-h) monthly lead frequency anomalies for the months of November to April in 2019/2020, respectively.

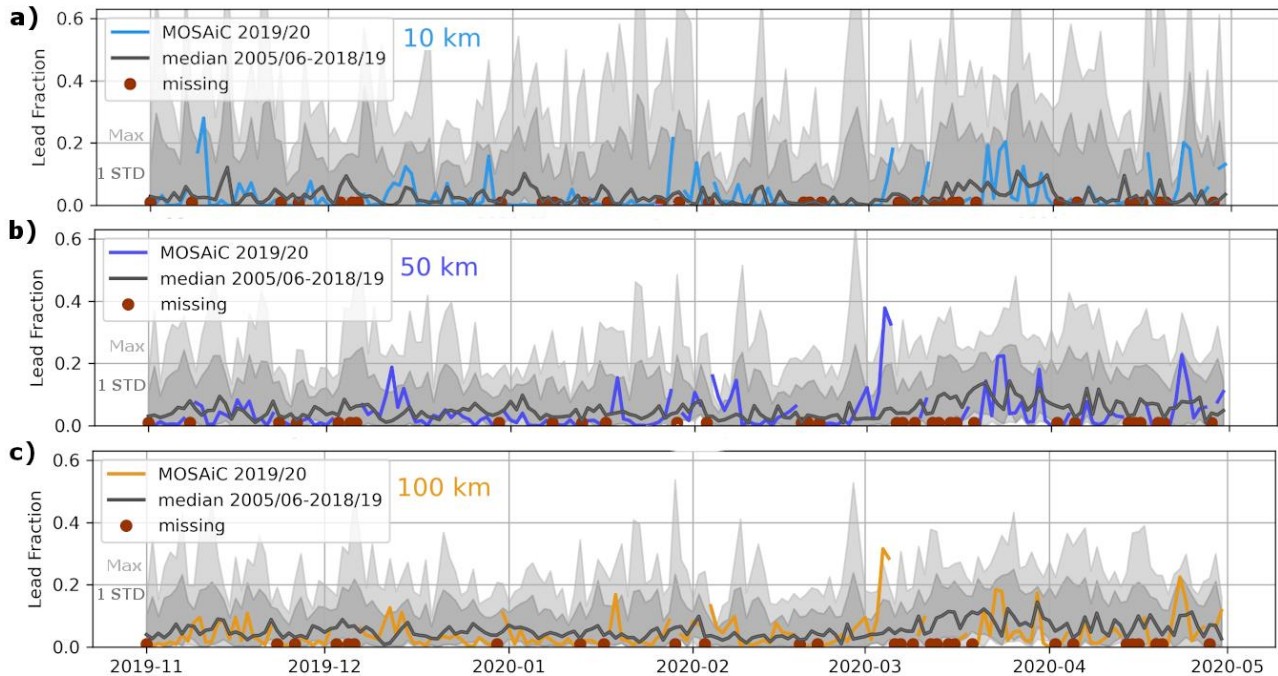

**Figure 16**: Temporal evolution of the lead fraction around the CO during MOSAiC for a) a radius of 10 km (light blue), b) 50 km (dark blue), and c) 100 km (orange), respectively. Maximum (light grey area), mean (black line) and one standard deviation (dark grey area) of lead fraction for the reference period 2005/06 to 2018/19 are shown for comparison.


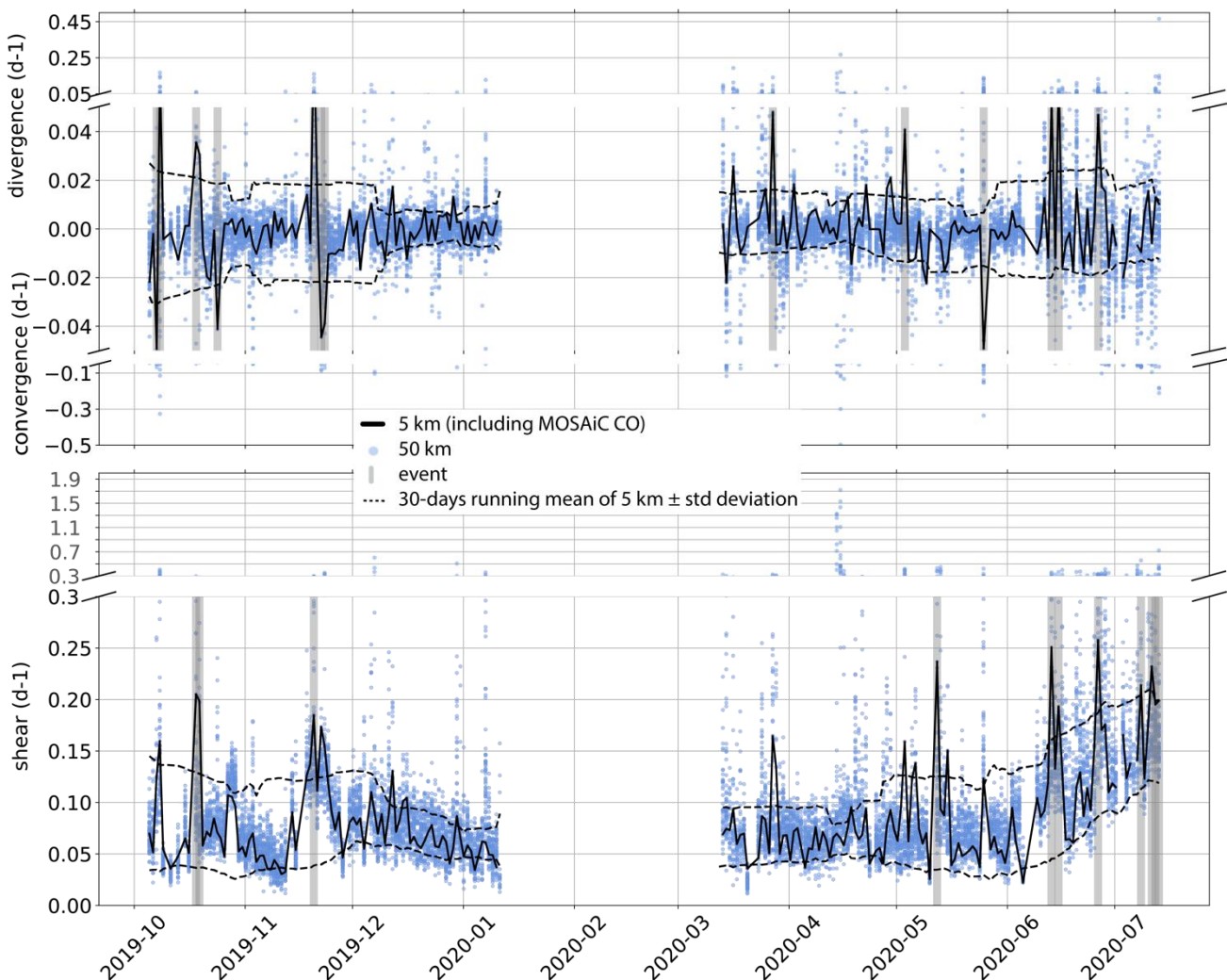

**Figure 17:** Time series of a) divergence, convergence and b) shear extracted from Sentinel-1 SAR scenes along the MOSAiC drift at different radii (5 km in black vs. 50 km in blue, compare Fig. 3) between October 5, 2019 and July 14, 2020. Strong deformation events with a magnitude of more than two standard deviations are marked by vertical, grey bars. Successive events might overlap and look like one event. The 30-days running mean ± standard deviation illustrates the seasonal variability. Please note the change in the y-axes spacing to better display larger deformation rates in spring and summer.

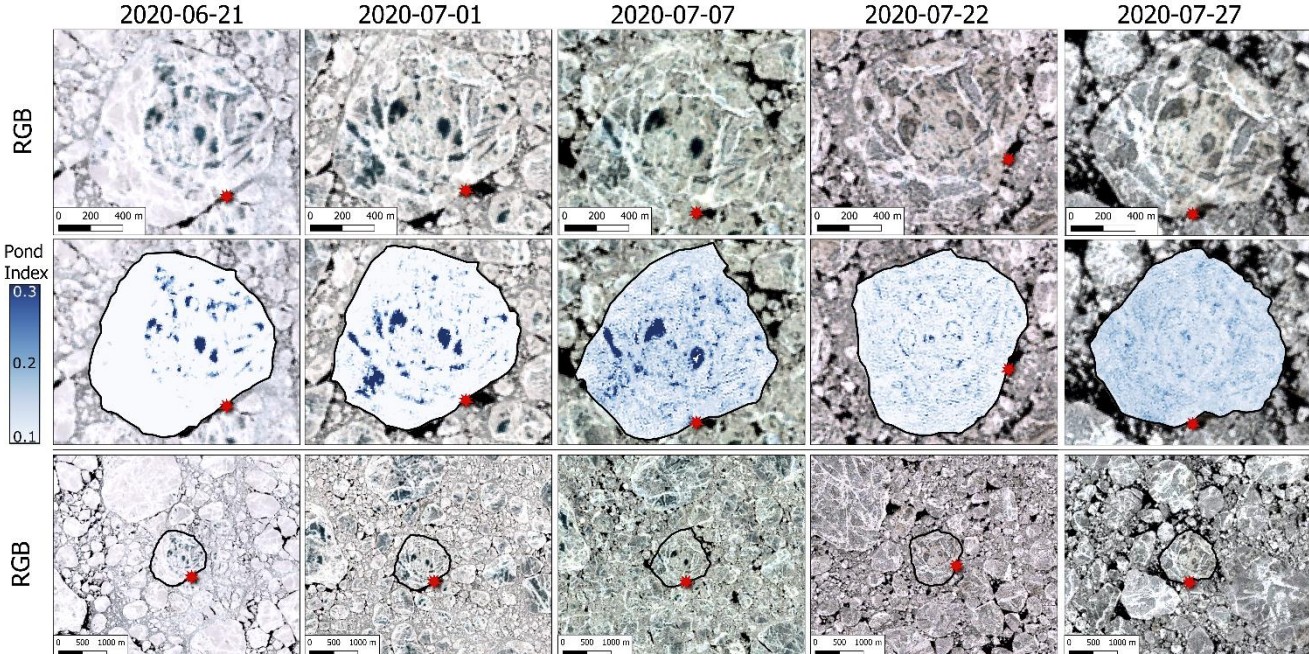

**Figure 18:** Upper panel: Sentinel-2 MSI True colour images of the MOSAiC floe obtained between 21July 21 and July 27, 2020. Central panel: Outline of the MOSAiC CO (black) and areas with classified melt ponds (pond index). Lower panel: True Colour image showing the larger surrounding of the MOSAiC floe (black outline) and the temporal evolution of melt pond on the surface of neighbouring floes. The Red star shows the position of *Polarstern* at the time of the satellite image acquisition.

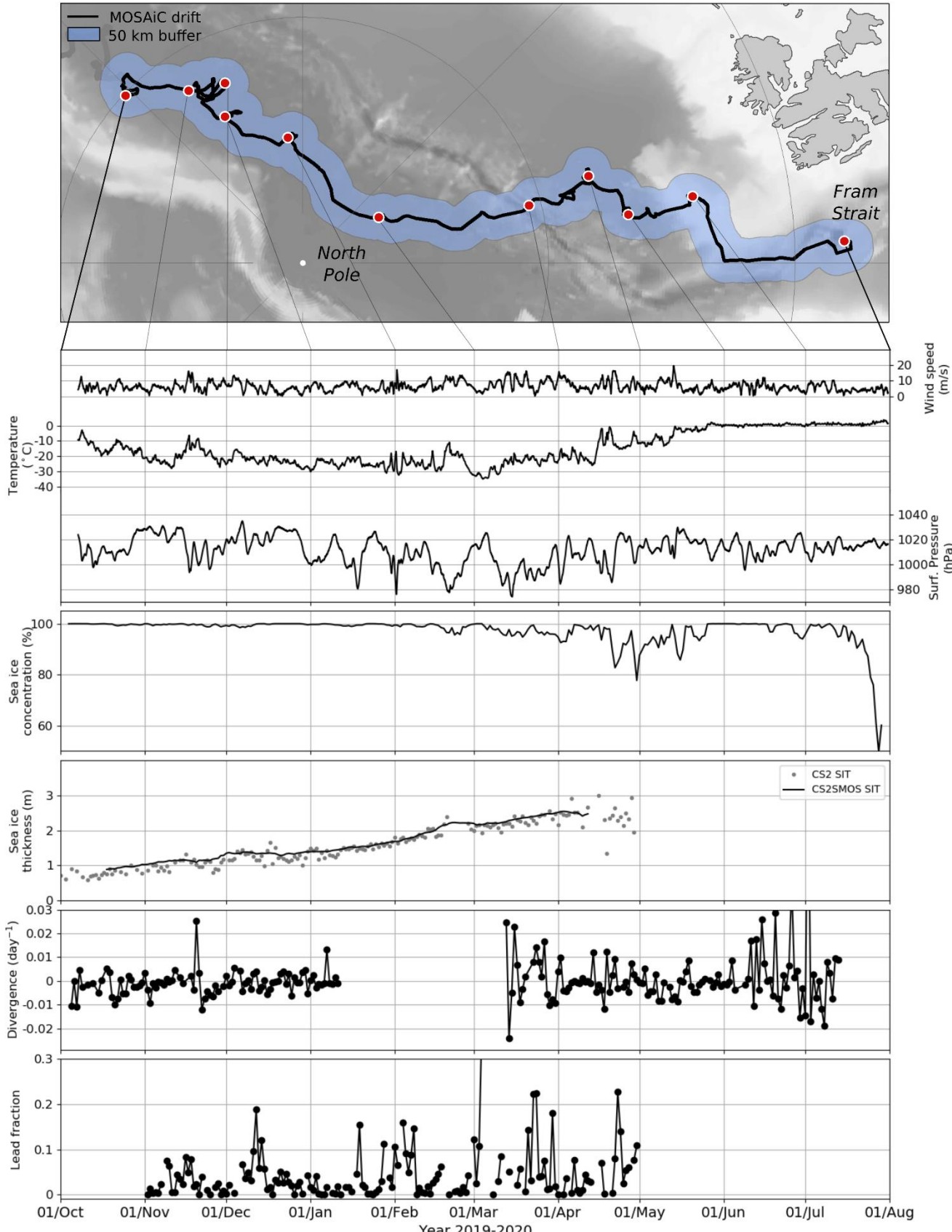

**Figure 19:** Summary of the atmospheric and ice conditions extracted from reanalysis and satellite data within a 50 km radius (top) along the MOSAiC CO drift track. Mind that the drop in sea ice concentration in April/May is apparent only due to wet snow (see Section 3.3).