# Peer review of "The MOSAiC drift from October 2019 to July 2020: Sea ice conditions from space and comparison with previous years"

_The Cryosphere, 2021_

## Author Comment (AC1)

**Reply to Reviewer 1:**

Dear Reviewer 1,

thank you very much for the comprehensive and helpful review of our manuscript. In the revised version we give a better insight into the uncertainties associated with the individual parameters and possible dependencies between the parameters. We "balanced" the discussion of the individual parameters and provide recommendations for research questions and upcoming studies. Please find below the detailed response to your comments.

Once again, we thank you for the time you have taken for this review.

With best regards,

The author team

**Summary:**

This paper reports on the physical sea-ice conditions (drift, coverage, leads and deformation, ice thickness and snow depth) as observed by satellite remote sensing sensors during the October through end-of-July drift of the MOSAiC observatory. The paper attempts to put these observations into context a) with the meteorological conditions - aka 2m air temperature, surface air pressure and near-surface wind speed - and b) with the conditions typical along the same but hypothetical drift track of the period 2005 until 2019. The paper is very descriptive but contains a wealth of different pieces of information. The paper provides intendedly little (critical) discussion or interpretation of the observations made. The paper provides little to no outlook about what next steps are already underway to i) better evaluate / improve the products shown or to ii) merge (some of) the products for improved understanding of geophysical interaction processes.

The paper gives access to the satellite remote sensing products obtained, covers an impressive set of parameters, and puts the observations made into a first wider context; it is hence a useful contribution to the scientific community.

I have two general comments (see below), followed by a number of specific comments and a few typos / editoral remarks.

**General Comments:**

GC1: My main general comment / concern about the content goes towards the imbalance in the degree of detail with respect to differences between the 50 km radius disc and the 100 km radius disc. For sea-ice thickness this is discussed at length - including hypothesis about why this difference of 4% is observed. For snow depth, one of the key parameters for freeboard-to-thickness conversion, this is not discussed. Neither are the differences in the lead fraction discussed nor the differences in the sea-ice concentration. I can understand that sea-ice thickness is an important parameter. However, in the light of the time series of the various parameters shown and in the light of the retrieval uncertainties in sea-ice thickness, I see other features that might be more worth to be put upfront / discussed in a bit more detail in such an overview paper.

Reply: We agree that the length of discussion is not completely balanced between the different parmeters. We improved that in the revised version. When results, however, are qualitatively similar between the different radii we mention that and only discuss the temporal development for the smalles radius in detail.

What explains the observation that despite an average 2m air temperature of -20degC throughout December sea-ice thickness did not increase? What explains the observation that apparently sea-ice thickness growth accelerated from January to February, staying at a high level in March, reaching a plateau first half of April despite continued -20degC 2m air temperatures? How different would results for the years 2005-2019 look if the parameters would have been retrieved along the reconstructed hypothetical drift tracks shown in Fig. 8?

Reply: The evolution of sea-ice thickness in regions with sizes that are defined by our two search radii is controlled to a significant degree by dynamic processes and not only the

thermodynamic sea ice growth. The behaviour of the mean sea-ice thickness referred to by the reviewer can partly be explained by the other remote sensing observations in this paper. The low increase, respectively slight decrease, in mean sea-ice thickness in November/December 2019 coincides with a strong divergence event that has added thin ice to the ice thickness distribution and thus effectively reducing mean ice thickness. The stagnation of mean ice thickness in April is a similar divergence event noticeable in a reduction of sea ice concentration and a widening of the sea-ice thickness distribution (SIT IQR and ICR).

We agree that we haven't described the connection between the different remote sensing data enough in the original version and added text to the discussion in the sea-ice thickness results section.

The mean ice thickness evolution of the previous years along the MOSAiC drift track has been displayed in Figure 13, showing that substantial changes of mean thickness can occur in the different years.

My believe is that this overview / first synthesis paper would benefit from a few more over-arching considerations and/or opening questions to be answered in forthcoming papers to be put at the end. I find the lack of opening doors for the work to come the weakest part of this manuscript.

Reply: We agree. We now provide key questions to address by future studies / outlook at the end of each chapter, in which the parameters are presented and discussed.

GC2: My second general comment is a follow-up to the degree of detail offered for the sea-ice thickness part. Given the retrieval uncertainties of sea-ice thickness it might be a very good idea to transparently communicate these uncertainties. Putting the observed L2P and/or CS2SMOS observations into context with earlier observations calls to at least note that uncertainties exist. I am not asking to discuss these at length. But comments like: "Observed snow depths (in situ!) agree well with Warren et al. (1999) snow depth climatology used in the sea-ice thickness retrieval ... " Do they? or "Sea-ice thickness retrieval at the border between first-year and multi-year ice is not trivial because the ice type determines values of key retrieval parameters snow depth and ice density ... " --> How solved here? or "SMOS sea-ice thickness products have yet received little evaluation for sea-ice conditions encountered during October/November ..."? or "The observed difference between observed and 10-year mean sea-ice thickness could potentially be explained by a mismatch between actual and climatological snow depth ... "? I guess all I ask for is to be a bit more specific at certain places and to be a bit more critical about the data merged together in this overview paper.

Reply: We agree that the discussion of sea ice thickness has been quite short and we have added information throughout sea-ice thickness description in the method and results sections. This manuscript summarizes (mainly) existing remote sensing products for the MOSAiC drift.

More details about the methods and their evaluation can be found in the cited references. We do however want to stress one point of our reasoning: Sea-ice thickness uncertainty described the uncertainty of the absolute sea-ice thickness value. The main uncertainty contributions for regional averages come from error sources such as freeboard biases, unknown snow load

or variable sea-ice density and not the actual noise of the sensor. The major error contributions have unknown correlation length scales but probably can be assumed as constant for our analysis here. Thus, especially the high resolution CS2 l2p data may be able to pick up local differences far smaller than the uncertainty of the absolute ice thickness values. We see the fact that the thickness differences between the two scales persist over the winter season as a confirmation. We thus have added text to the sea-ice thickness sections both to discuss the uncertainty of the data products as well as a discussion of our reasoning.

We also discussed the impact of sea-ice thickness uncertainty when comparing data between different years. Here we are now referring to results from a recently published study (Mallett el al., 2021) that indicates both stronger variability and trends when using more realistic inputs, in this case snow load, to sea-ice thickness retrieval chains.

And on the subject of verifying parametrization in the SIT retrieval chains with MOSAiC field data: We choose not to do such a comparison at this point because we want to document and discuss the current state of remote sensing data. Such an analysis needs to take the full wealth of MOSAiC field data into account, which is beyond the scope of this paper. But it is a good point and we added it to the outlook paragraphs (see your last GC).

**Specific Comments:**

Title: Any reason why you did not choose to write "physical sea-ice conditions"?

Reply: We changed it to "The MOSAiC drift from October to July: Sea ice conditions from space and comparison with previous years". We decided to not add "physical" to keep the title concise but we added "physical sea ice" in the first sentence of the abstract.

L26: I suggest to note what you mean by "previous years" and later on, on which years the climatological mean is based.

Reply: We agree. Reference period was missing in the abstract, conclusion and in a few other sentences. We now ensure that the term "previous" is defined, when it is used in the text (e.g. past 14 years, period 2005/2006 - 2018/2019, etc.)

L35: I suggest to check how safe it is to come up with such a detailed number when speaking about the ice thickness which is known to have a substantial uncertainty. Is 4% within the retrieval noise? This refers back to GC1 and GC2.

Reply: We have revised the wording in the abstract to acknowledge the small degree of the thickness difference. "*Here we find that sea-ice (within 50 km radius) of the CO was thinner than sea-ice within a 100 km radius by a small but consistent factor (4%) for successive monthly averages.*"

We do agree that the thickness difference is fairly small and with a magnitude of a few centimeters well below typical sea-ice thickness uncertainty estimates for radar altimetry at these scales. It should however also be noted that the thickness uncertainty applies to the absolute value and not necessarily local thickness differences. Averaging along-track thickness values at the scales of the two search radii will reduce retrieval noise substantially. The remaining thickness uncertainty rather depends on components such as the snow load variations between the year that are likely to have far larger correlation lengths. Thus, the altimeter is able to pick up small differences far below the absolute uncertainties of these values. We take the fact that the thickness difference is consistently present in all month as a confirmation that this is not caused by retrieval noise.

We do however agree that the accuracy and precision of the thickness information needs an expanded description. Please also see the changes made to the description of thickness uncertainty made in response to GC1 and other specific comments.

An additional point to make is that the thickness difference would likely be larger if the thickness averages of the 50km search radius would not be a subset of those from the 100km search radius.

L61: I suggest to provide a motivation for the length and years included in the reference period. It is neither a classical climatological period - aka 30 years in mid-latitudes, nor does the length resemble any other common period.

Reply: The reference period was chosen such that there is a maximum overlap of the data products used (e.g. lead product is only available after 2004). Moreover, Krumpen et al. 2019

shows that after 2004, there is some sort of regime shift in the TPD, as the survival rate of sea ice formed on the Siberian shelf seas decreases thereafter. We now provide a short note in the text.

L94: Since you used the version 4 product you could in principle also cite the TC paper from 2020.

Reply: Good point. We replaced the reference with Tschudi et al., 2020

L96: Was this data set also assessed in the work of Sumata et al.? In this case I recommend to also include a respective reference.

Reply: We agree. The Sumata et al. (2014) work is not exactly about IceTrack and based on NSIDC version 3 data. We now refer to Krumpen et al. 2019, which does not include version 4 either, but provides a first indication of reduced tracking performance in Fram Strait area. Please compare changes made in the manuscript.

L98: CERSAT summer-time ice motion is missing for good reason. I guess it would be fair to mention that in June/July NSIDC ice motion is nothing else than based on NCEP wind - supported with buoy motion (if present).

Reply: We agree, NSIDC summer motion estimates are far from perfect and close to NCEP wind. However, to discuss this topic properly, with all its background, a rather extensive explanation would be necessary. E.g. one would also have to describe why the OSISAF product delivers good data in summer, unlike other products. We believe that this discussion would lead too far and would not improve the manuscript in the long term. We have therefore made no change and rely on the references provided in the manuscript for more in-depth information.

L99: "Sea ice is traced forward ..." It might make sense for clarity that the sea ice at the starting position of the CO is traced forward in time. You are not looking at the entire Arctic Ocean. It might also make sense to be just a little bit more specific in terms of how this tracking is applied. A) How large is the parcel traced forward? B) Do you interpolate the gridded ice motion information onto the location of the ice parcel? How?

Reply: Yes, this information was missing. We added "at the starting position of the CO". The tracking procedure itself is described in detail in Krumpen et al. 2019 and Krumpen et al. 2020 and this description is rather meant to be a summary. We hope that providing a reference for readers interested in more details is sufficient.

With respect to the parcel tracing: The parcel is independent of the resolution, since we apply a lagrangian approach and interpolate (as you assume correctly) the gridded ice motion information onto the location of the virtual buoy/CO position. A weighted mean is used for this. We tested different interpolation methods (e.g. nearest neighbour), but differences are rather small.

L99: Why 1996 as the starting year here?

Reply: The starting year was indeed chosen somewhat arbitrarily. Actually, we can compute, with the given motion data, drift trajectories back to 1992 (compare Krumpen et al. 2019). However, this was overlooked. We hope that the chosen period, even if shorter and not properly justifiable, is okay.

L100: "if ice concentration at" --> "if the sea-ice concentration at"

Reply: Changed, thanks.

Question: Which sea-ice concentration product do you use? What is its grid resolution? Is it sufficient that the sea-ice concentration drops below 50% once or does it need to be a number of consecutive days?

Reply: This information was missing in the description. We now state: "Sea-ice concentration along the trajectories is provided by CERSAT and based on 85 GHz SSM/I brightness temperatures (6.25 km resolution)".

Tracking will stop immediately if ice concentration drops below threshold.

L106 / L110: I am confused with respect to the Fig. 2 and Fig S1 in the supplementary material. These appear to be identical. It should be either or. This applies to some of the subsequent figures as well.

Reply: Yes, sorry. Originally, the manuscript came with a supplement, where most of the figures from chapter 2 were made available. This was changed prior submission, however, it looks like we did not remove all links to the supplement. Corrected. Same comment was made by Reviewer 2.

L124/125: You write about "uncertainties" and "bias". I suggest to make sure that the reader understands whether your "uncertainties" are in fact the bias (or the accuracy) or whether you also include precision (or the standard devation around the true SIC value) into this expression. If there is, e.g., a notorious underestimation of SIC during summer due to melt conditions, i.e. a negative SIC bias, then there is possibly little chance to reduce the bias because neighboring grid cells are not independent and melt processes are typically larger scale phenomena.

Reply: You are right. We made a clearer distinction between accuracy and precision and mention a potential low SIC bias during summer.

L128: "multi-parameter retrieval" --> that is based on data of which sensor?

Reply: It is also based on AMSR2 observations but uses all frequency channels and retrieves seven surface and atmospheric parameters simultaneously (by a forward model inversion), which, in this case, seems to mitigate some of the wrong SIC retrievals. Added this information to the text.

L145: "at least 50 data points" --> Perhaps add: "within the specified search radii".

Reply: AddedL146: "as very few orbits ..." --> You could perhaps add that because of this there won't be any direct inter-comparison between SIT measurements taken at the CO and CS-2 observations?

Reply: Added. *"For the same reason we also refrain from comparing short segments of L2p data to local observations on the CO, not only because of the lower temporal coverage but also because the retrieval noise in the l2p SIT data will dominate on the scale of the local SIT observations."*

L149: "also north ... 88°N" --> While it is of course good to have this opportunity to "fill" the pole hole it might be fair to state that any SIT values north of 88degN that you retrieve and show along the track of the CO are basically the result of an informed extrapolation. Any gradients in SIT that might occur north of the latitudinal limit are not represented.

Reply: A step in the optimal interpolation of the CryoSat-2/SMOS data is to compute the correlation length scale of sea-ice thickness in areas where input data is available. In the central Arctic this correlation lengths scales are typically estimated as 150km and Polarstern did not drift fully into the pole hole further than this value. While it is true that the satellites are oblivious to any thickness gradients confined to the pole hole, gradients of a larger scale can be approximated by the optimal interpolation.

We clarified this in the manuscript:

*"CS2SMOS thickness estimates at the CO position during the short period when Polarstern drifted north of 88N are thus based on a spatial extension of SIT gradients measured at the CS2 orbit limit."*

L159: At the end of this subsection I note that you provided information about the uncertainty for the sea-ice concentration product and in some sense also for the drift data (ok, for the drift track) but not yet for the SIT data used. I guess it wouldn't hurt to write a few lines about SIT retrieval uncertainty here - ideally you'd also state already the issue that a merged snow depth products needs to be used which is partly based on an (outdated) climatology which prevents adequate mapping of the inter-annual SIT variability.

Reply: Accepted. We added a sentence: "...CS2SMOS thickness estimates at the CO position during the short period when Polarstern drifted north of 88N are thus based on a spatial extension of SIT gradients measured at the CS2 orbit limit."

L167: Where does the ice type information come from?

Reply: We added reference for the ice type retrieval to the text (Ye et al. (2016a,b), data can be downloaded from https://seaice.uni-bremen.de/multiyear-ice-concentration)

L168/169: "estimate of the uncertainty" --> This sounds perfect. Please then provide an estimate of the respective values. Perhaps a range would do it? Or some percentage values?

Reply: That is a good suggestion. We added the uncertainty range (5-10 cm and a specifically for MOSAiC calculated uncertainty of 8 cm) to the text.

L169/170: "Snow depth currently ..." --> Perhaps, for similar studies in the future, it might make sense to revive the SMOS-based snow depth estimation approach by Maass et al. ?

Reply: Yes, the SMOS or the ICESat-2 plus CryoSat-2 snow depth products would be good alternatives to also look at. We added that to the outlook section (see your GC1). Indeed, we do not discuss all available satellite datasets (also not for other variables like ice concentration or thickness) but selected one for each sea ice property. Admittedly mainly based on preferences of the authors. Thus, future studies should compare different datasets and then also use the in-situ data collected during MOSAiC to make an evaluation. That is not yet the purpose of this study (which is mainly an overview and comparison to climatology).

L172: "Here we present ..." --> Hmmm ... given the fact that the grid resolution of the snow depth product is 25 km ... how do you realize this? Isn't this basically one grid cell?

Reply: Yes, what we call 12.5 km radius here is 1 grid cell of 25x25 km^2 size. Clarified in the text. To make comparisons between different datasets we refer to different length scales by "radii" in a consistent manner to make the text better comprehendible. For gridded datasets these are not necessarily circles. As datasets are compared only qualitatively and the temporal development for data averaged at different radii usually is very similar this is not introducing a significant error.

L177: How is a lead frequency defined at daily temporal resolution (L184)? Could it be that lead frequency is a monthly product? In addition, what does a frequency of, e.g. 0.1 mean? Does this mean that a lead was identified at the respective location on about 3 days within the respective month?

Reply: As stated in the text, the lead frequency is a temporally integrated product and is thus only given as monthly values based on daily data. As such, a frequency of 0.1 means that in 1 out of 10 cases (days) a lead was present during the referenced time period (here: months).

A related question: Does it make a difference whether a lead occurs at a specific location on 3 consecutive days or on 3 arbitraily distributed days within the respective month?

Reply: I would say that this depends on the question that is being asked. It probably does when one wants to look into specific case studies (spanning only days) and associated heat loss investigations. For a more holistic view that tries to give a general overview (as here) I do not think it does.

L186: Is 10 km the grid resolution (pixel size) of the lead product? Please add this information.

Reply: The daily lead product has a spatial resolution of 1 km. The sentence in L186 was misleading and changed accordingly. Thanks.

L192: Sea-ice drift and deformation fields are offered at the same spatial resolution? Does this imply that the grids (the one where the motion is provided and the one where the deformation is provided) are shifted with respect to each other by half a grid cell?

Reply: Yes. We considered for every derivative a set of four grid cells and located the result to the center of the four grid cells. Moving with a step width of one grid cell, the grid spacing of

the deformation fields remained the same as the one of the drift fields. We direct the interested reader to the publication von Albedyll et al. (2021) to find those details.

L202: "... standard deviations." --> This is the standard deviation of the deformation parameter (i.e. convergence or divergence or shear) averaged for each individual 5km-radius-circle areas? Or is this refering to the standard deviation derived over the 61 circle array? Or, to ask it in a different way: occurs the definition of a strong deformation event at the scale of the 5km-radius circles or at the scale of the gridded deformation, i.e. 1.4 km?

Reply: Thanks for the question that made us realize that important details were missing in the text. The classification of "strong deformation events" was made based on the 5km averages of the CO time series. We moved this part and specify in the text: *"Exceptionally strong deformation events are defined as events with a magnitude exceeding two standard deviations of the 5 km averages time series."*

L220: I suggest to add the grid resolution of the ERA5 data used by you.

Reply: The ERA5 data are on a 0.25° grid. We have added this information.

L236/Fig. 6: I suggest to avoid that data of the surface air pressure and data of the 2m air temperature overlap each other for the individual years. I would also make sure that the respective y-axes cover the full range of every parameter shown.

Reply: We have followed this suggestion.

L238: "+10°C" --> Do you mean that air temperatures as high as +10degC were observed? Or do you refer to the temperature anomaly? As a meteorologist I'd always see the unit Kelvin used for a temperature anomaly to avoid misinterpretations. Hence: "up to 10K, not shown"

Reply: Indeed we are talking about anomalies here. As suggested, we have changed the units to K for temperature anomalies to avoid misunderstandings.

L243: Sloppy wording. Please specify what you mean by "stormy conditions".

Reply: We have changed "Stormy conditions" to "High wind speeds".

L246/Fig. 7: I would make sure that the respective y-axes cover the full range of every parameter shown.

Reply:  We have followed this suggestion.

L248: "the raw on-board ..." --> Why is this? Because of the different anemometer heigts? Didn't you correct the wind speed measurements to 10 m height? If not why not? Since you have mostly stable conditions it might be a straightforward thing to do and then you might be able to make a more quantitative statement about the ERA5 wind speed quality. How about wind direction?

Reply: We have now specified the exact measurement heights, which is 39m for the wind. We have not applied a height correction and argue that this is not required for the qualitative

comparison we are presenting here. We are now explicitly stating that a more stringent evaluation of reanalysis data with MOSAiC in-situ measurements – of the ship and numerous other sensors across the CO and DN – will be treated elsewhere.

L249: change unit to K.

Reply:  We have followed this suggestion.

L249-251: "especially taking into ..." --> I suggest to be a bit more explicit here. I'd say there are two effects to take into acount. The ship heats the air aloft and might cause higher air temperatures at the measurement height than in the free atmosphere. And at the same the measurements are taken substantially further away from the surface and hence the near-surface inversion of the temperature might not be captured well.

Reply: We have added the following sentence to make this more explicit: "Note that the true 2 m temperature bias might be even larger because the ship air temperatures might be high-biased due to (i) local heat sources and (ii) higher temperatures at the measurement height of 29 m compared to 2 m in typical cases of near-surface inversion."

L253/253: "Given the fact ..." --> This is one way how to close this paragraph. Another way would be to note that during MOSAiC a large set of air-temperature measurements along vertical profiles was taken which is (presumably) going to be used in a forthcoming study to more quantitatively assess ERA5 2m air-temperature data against the MOSAiC observations - including the ship-based data.

Reply: We have added the following sentence to the corresponding paragraph in Section 2.8 (Reanalysis and ship data): "The ship measurements are however taken at non-standard heights (wind: 39m; temperature: 29m; pressure: 16m, reduced to sea level) so the evaluation is rather qualitative. More stringent comparisons of MOSAiC in-situ meteorological observations, not just from the ship but from a large number of sensors across the CO and DN, are beyond the scope of this paper but will be conducted elsewhere."

L265/266: "The interannual variability ..." --> How you know? Can you exclude that ice thickness / ice-surface structure due to deformation doesn't have a similarly large effect?

Reply: Sentence was removed

L272: "97%" --> This number is based on which data set?

Reply: The 89 GHz SIC, i.e., contains the underestimation in April. Information added to the text.

L279: "was with 99.5%" --> A good moment to cite papers such as Kwok, 2002 or Andersen et al., 2007, referring to the high-concentration Arctic average sea-ice concentration values.

Reply: Thank you, we agree and added the reference to the text.

L282/283: I possibly missed this: How did you compute the distance to the ice edge? What was your reference point?

Reply: You are right, we had not described that. The information how the distance to ice edge is calculated are now added to the data section 2.2:

*"Based on the 89 GHz sea ice concentration dataset we also calculate the closest distance from the MOSAiC CO to the ice edge: to remove small openings in the ice we first smooth the ice concentration dataset by convolution with a 4×4 (25 km) grid cell kernel, then the distances from the CO grid cell to all grid cells with zero ice concentration are calculated and the shortest distance is selected as distance to the ice edge."*

L285: "stayed higher" --> Please be more specific. Was it 99% ... 95% ...?

Reply: 97% based on the visual observations from the bridge. We added "higher than 95%" to the text as the uncertainty of these visual estimates is quite large.

L286/287: "We can sea ... Fig. 6" --> I recommend to mark this period in Fig. 6 for better clarity.

Reply: If this paper only would be about this April event, we agree that this period should be marked. However, the Fig. 6 meteorological time series is used for different purposes and we think it would get to crowded if we would mark this event and other events.

I am wondering what the effect of the increase in wind speed might have had on the physical snow and sea-ice properties relevant for its microwave remote sensing?

Reply: This is an interesting question. The wind can create snow slap layers and wind crusts with different snow density, also snow surface roughness can change. We are currentyl working on a specific microwave remote sensing paper for this event, which also will include some of the on-ice MW radiometers and other in-situ measurements. A more detailed discussion of this is beyond the scope of this paper but we mention this in the outlook now.

L287/288: "liquid water ..." --> Please provide 1-2 references to back up your statement about this important process.

Reply: We reformulated this now more specific and added references:

*"The warming induces strong temperature gradients and increase vapor fluxes in the snow, which can cause stronger snow metamorphism and significantly change the snow permittivity already at above −5°C snow temperatures (Mätzler, 1992). Also, liquid water content can already increase at temperatures slightly below 0°C and already small liquid water fractions of, e.g., 2% strongly change the microwave loss in the snow (Hallikainen, 1986). Refreeze after the warming event can cause ice lenses in the snow. Such events already previously were observed to influence microwave properties and penetration (e.g., King et al., 2018). On 19th April slight drizzle was observed, which likely refroze on the snow afterwards."*

L288-290: "These surfaces ... used here." --> It might not hurt to also back up these statements by a reference.

Reply: We added a reference that discusses that: Lu, J., G. Heygster, & G. Spreen (2018). Atmospheric Correction of Sea Ice Concentration Retrieval for 89 GHz AMSR-E Observations. IEEE J-STARS., 11(5), 1442–1457. doi:10.1109/JSTARS.2018.2805193

L293: "which takes" --> I know that work and I suggest to be more careful in the writing: "which takes" --> "which attempts to take"

Reply: Agreed, we reformulated that more cautious: "... which attempts to take such effects into account (specifically the atmospheric influence) and in our case is in better agreement with the ship-based observations."

L294: Fig. 9: What I am missing at this point is that Fig. 9 indicates that sea-ice concentrations dropped below 90% in a number of previous years - just to put your observation into a wider temporal context.

Reply: Agreed, we mention now in the text that also occasionally other years showed low SIC during winter.

L296: Just as a reminder: I am not sure you wrote how you derived that distance. And, of course, given the way you use this information in this manuscript, I am wondering why you did not exclude the Russian marginal seas after freeze-up rightaway.

Reply: See comment above. The description of the ice edge calculation is now included. If the ice edge can also reside in larger polynyas is a matter of choice. We kept it as it. Also larger open water areas in polynyas could have impact on atmospheric moisture transports and heating of air masses.

Figure 12: I suggest to add to the caption that 50km SIT estimates are missing in case there were not enough available L2P data within the 50km radius disc.

Reply: Added

Table 1: I suggest to add a column in Table 1 which provides the average number of valid data that was used to compute the mean, IQR and IDR.

Reply: Table 1 provides only monthly values and the number of valid data is varying on a daily basis. As an alternative we have provided information on the number of L2P data points, as well as CS2SMOS data points in the method section.

 *"The number of CS2 l2p data points for the 50km (100km) search radii varies from approximately 50 (300) at lower latitude to approximately 900 (2000) close to the maximum orbit coverage of CS2 at 88N.."*

*The number of CS2SMOS SIT observations selected may depends on the position of the CO relative to local grid cell coordinates. The number varies between 10 and 14 grid cells for the 50 km and between 47 and 52 for the 100 km search radius."*

L304: "period between ..." --> I assume that this is the period when the 100km radius of the disc centered at the CO first / last intersects with the observational data gap at the pole? Or did you in fact use L2P CS2 observations until the CO traveled across that border? In that case there is a phase before / after where the number of valid L2P decreases / increases. How was this solved in detail? In a way I don't fully understand your writing in the context of Fig. 12 ... showing continuous 100 km sea-ice thickness data.

Reply: We search for CS2 L2P data points for all days of the drift including when Polarstern drifted north of 88N degrees. This is done to be consistent since also for other days there might be CryoSat-2 orbit coverage at the edge of the search area. As described in the document we exclude days where there were less than 50 CS2 L2P data points in the search area. Thus, it is likely that at 100km scale there are still CS2 L2P data points falling into the 100 km search radius, even if the center of search radius (Polarstern noon position) is north of 88N. We have updated the text in the manuscript and the caption of figure 12 to describe this better.

L311: "was consistently" --> "was, on average, consistently" ... just to better comply with the observation that there are quite some locations where the 100km radius disc SIT is smaller than the 50km radius disc one.

Reply: Changed

L323-325: I have difficulties to understand this. I would have thought that, as you wrote, the L2P data provide an accurate "point" measurement, or better, a suite of such measurements ... covering the full sea-ice thickness range down to, say 0.2 m. When there is not thick enough ice, there won't be any measurement. Hence, thin ice is excluded and an average might in fact be biased high. CS2SMOS in contrast fills that gap by including thin ice thickness measurements, hence adding the thin tail to the ITD; because of this I'd expect that CS2SMOS is providing thinner average sea-ice thickness values than L2P CS2 data. Why is it the other way round here - according to your writing?

Reply: The difference between CS2 L2P and CS2SMOS is mostly the spatial scale. CS2 L2P solely depends on the respective footprint of individual radar echoes, while CS2SMOS is based on optimal interpolation of CS2 L2P data and gridded SMOS data with an interpolation window. Thus, if a local ice thickness minimum exists, the L2P data is better able to pick it up for reasons of spatial resolution only CS2SMOS depends on observations of a far larger influence region. CS2 L2P data from very thin ice exists in the processor. In fact, we also allow negative thicknesses to not bias the noise distribution at the lower end of the ITD.

L333-335: "The monthly sea-ice thickness ..." --> How do these differences relate to A) the retrieval uncertainty or error of the CS2SMOS product and B) to the standard deviation of the multi-annual average CS2SMOS sea-ice thickness?

Reply: Please see our response to GC2

L345: "good agreement ..." --> This is a rather unspecific statement. Which errors? What are the "expected uncertainties"?

Reply: Okay, we added the uncertainty of 5 cm for the satellite dataset and mention further evaluation of the dataset by in-situ observations in the outlook paragraph at the end of the subsection.

L348: "Only after ... " --> Would it be possible to show precipitation in Fig. 6 along with the other three parameters? This would aid greatly in the credibility of the statements made in this sub-section.

Reply: We though about adding ERA5 precipitation rates to Figure 6. However, estimates are uncertain and would require a proper discussion of the associated uncertainties. This topic will be addressed by future studies.

L349-353: "After ... earlier." --> I'd say that these lines are not well backed up by in-situ observations and/or ERA5 data. I suggest to remove them. Interaction between snow properties and microwaves in the presence of snowfall (or other precipitation like freezing rain) and in combination with substantial variations in the near-surface air temperuture as shown in Fig. 6 are very complex. There have been studies in the past demonstrating that adding a few centimeters of snow can have a profound influence on the brightness temperature. In addition, snow metamorphism (aka a change in grain size) might also have played a larger role than snow compaction.

My suggestion would be that you condense this paragraph to what we can see in the passive microwave snow depth estimates, ideally say something more specific about the in-situ observations and, which I would find of utmost importance given the relevance of snow depth for the CS2 sea-ice thickness retrieval, relate these observations to the Warren et al. (1999) snow depth climatology.

At the end you could then state that an immense measurement program took place during MOSAiC to make a step change in our understanding how microwaves react to varying snow conditions on sea ice.

Reply: Thank you for this constructive comment, which we are happy to follow. The microwave interaction with snow and ice is now discussed in some detail in the sea ice concentration section 3.3. Therefore, we do not repeat it here. The paragraph now reads:

*"This is in agreement but potentially a bit lower than the detected snowfall by several sensors in the MOSAiC CO (about 10–20 mm SWE, i.e., approx. 4–8 cm snow depth; Wagner et al., 2021). However, also the Wagner et al. (2021) study shows that snowfall does not always directly relates to snow depth increases because lateral snow redistribution plays a significant role. Future studies will evaluate the satellite snow depth in more detail based on the extensive snow measurements taken during MOSAiC. Based on that we likely will be able to extend the satellite snow depth time series to the beginning of the drift in October.*

*The satellite AMSR-E/2 March/April snow depth of 22 cm is significantly lower than the snow climatology from Warren et al. (1999) for the years 1954 to 1991. For this the March/April snow depth for the MOSAiC region would have been between 35 and 39 cm, i.e., 60% to 80% higher than during MOSAiC and the whole AMSR-E/2 time period 2005 to 2019 (green line in Fig. 14). Thus, we observe a strong reduction in snow depth for the MOSAiC region compared to previous decades. This also has implication for ice thickness retrievals from satellite altimeters, where the Warren snow depth climatology often is used for the freeboard to ice thickness conversion (Section 2.3 and e.g., Ricker et al., 2014).*

*Here we only present one satellite-based snow depth product. Future studies will compare our snow depth retrievals from the AMSR-E/2 microwave radiometers with snow depth from combined CryoSat-2 and ICESat-2 measurements (Kwok et al., 2020) and snow depth from SMOS (Maaß et al., 2013)."*

L357: "The lead frequency is ..." --> You might want to put this information into the respective subsection of section 2.

Reply: Text is moved as suggested

L371-373: "Only in March ..." --> You could relate this observation also to the respective figure showing the anomalies in the atmospheric circulation which for March and April agree well with your results.

Reply: Relation is added.

L384/385: "It is striking ..." --> Why? Perhaps remove the "striking" part and simply state that these events were not necessarily accompanied by a decrease in sea-ice concentration because of (... timing of lead opening vs. clear-sky image acquisition / near-surface air temperature determining freeze-over / intermittent change in wind direction - aka lead closing ...)

Reply: That's right. Thanks. Text is modified accordingly. Timing of lead opening vs. clear-sky image acquisition will, however, be of minor importance because the MODIS data are daily averages as well.

L390-392: "Moreover, we find ..." --> I am not sure I would compare the results of the entire MOSAiC drift with the results cited here because the geographical region the latter are representative of correspond to MOSAiC from May onwards. Perhaps you could relax your statement into that direction. In addition I recommend to add that the Oikkonen et al (2017) results are from the N-ICE2015 drift campaign.

Reply: Thanks for pointing out that more background information is needed. We rephrased this in the text: *"Despite the different geographical regions, we find that mean shear and divergence of 8 % d-1 and 2 % d-1 along the MOSAiC drift track are in good agreement with deformation rates obtained from a ship-radar North of Svalbard during the N-ICE2015 drift campaign (Oikkonen et al., 2017)."*

L403/Fig. 17: Since it is not obvious from Fig. 17 that 60% of the events took place in Oct./Nov. you could add to the caption of Fig. 17 that vertical bars of successive events might overlap and look like one event ... or whatever you like to avoid the impression that 60% is a wrong number.

Reply: Thanks for the suggestion. We have added this statement to the figure caption.

L404/405: It would have been really cool to be able to delineate this event also in Fig. 17. Why is it missing therein?

Reply: It was previously not visible because we limited the y-axis to display better the seasonal variability. Please have a look at the shear time series below without y-axis limits where the event on April 14-17 is well visible. Further please note that the event did not affect the 5km CO circle. Therefore, it was not noted as "strong deformation event" and not marked with a gray bar in Fig. 17. To be able to display seasonal variability AND all data points, we have produced a new version of Fig. 17 with broken y-axes (see manuscript).

[Figure]

L411-417: How likely is it that the months long action on the floe has resulted in snow property changes that triggered faster melt? Also, if I am not mistaken, then the sediment argument only can come into place when the snow cover is completly gone.

Reply: Large parts of the floe were not accessible because they had been declared in advance as validation sides for satellite measurements. Other parts were only accessible via pre-defined tracks. Hence, only areas in the direct vicinity of the ship may have experienced snow property changes that may have triggered faster melt. Naturally occuring deformations of the floe (i.e. ridges) are thus expected to be the dominant processes for the meltwater / melt pond distribution on the MOSAiC floe.

Sediments and even small pebbles and bivalves were found at the surface of the MOSAiC floe when the snow was gone (already in July). However, it is possible that the sediments had a preconditioning effect on the melting processes, because they lead to an increase in radiative heat due to the increased absorption compared to snow, which can then lead to enhanced melting. Since the processes that promoted early melting have not been studied in sufficient detail, we have deliberately left this part vague in the manuscript.

However, following your suggestion we provide key questions at the end of the chapter that may be addressed by future studies: *"The unusual temporal and spatial evolution of melt ponds on the MOSAiC floe compared to the surrounding floes raises the question of what processes preconditioned the early melt. More specifically, what role did the heavily deformed area play in the formation of melt ponds and to what extent did the presence of sediments accelerate melting processes?"*

L445/446: "Significant changes ..." --> I suggest to make clear that these changes are i) artificial as provided by the sea-ice concentration product and ii) are concomitant with elevated but still below-freezing temperatures. The way written could easily be misinterpreted towards: "high aire temperatures melted sea ice" by a non-expert.

Reply: Thank you, we agree and have reformulated that bullet point

L457: As the lead time series terminates end of April I suggest to delete "and summer".

Reply: We agree and removed "and summer".

**Typos / Editoral remarks:**

GC0: This is a general comment about the editing. I am wondering why the authors decided to write in mixed passive and active voice. I would find the paper easier to read if you'd have used active voice througout.

Reply: This is a valid point and we had many discussions about it. We have used active voice where possible.

L30: month --> months

Reply: Thanks, changed

L217: "temperature" --> I suggest to add "air" ; it might make sense in general to make clear that you are talking about air temperature and air pressure in the following (e.g. in L226) and hence be more specific in your writing.

Reply: thanks, changed

L217: "derived" or simply "taken"?

Reply: Corrected

L264: "westerly" --> Perhaps better: "westward"?

Reply: Changed

L276: "will" --> "we"

Reply: Changed

L280: "than the" --> "than during the"

Reply: Changed

L341: I note that the heading of the sub-section says: "Snow depth" but here you write about snow thickness. I suggest to use one term.

Reply: Changed

L380-382: "However ... dots." I suggest to move these two sentences further up in this paragraph, right behind the first sentence in L375.

Reply: Yes, it improves readability. Changed.

---

## Author Comment (AC2)

**Response to Reviewer 2:**

Dear Reviewer 2,

we are grateful for your constructive suggestions and positive assessment of this manuscript. In the attached file, we respond to all comments. In the revised version we give a better insight into the uncertainties associated with the individual parameters and provide a table that lists investigated parameters and their spatial and temporal resolution.

Once again, we thank you for the time you have taken for this review.

With best regards,

The author team

**Summary**

This paper provides an overview of the sea ice and atmospheric conditions along the Polarstern drift track during the first phase of MOSAiC. An impressive number of satellite data products are used to achieve this. While the manuscript isn't the most exciting from a scientific perspective, I appreciate that it wasn't intended to be. It will act as great reference material for anyone interested in working with MOSAiC data and I would be very pleased to see it published. However, I have some comments to be addressed first.

**General Comment**

The manuscript contains a huge amount of information, and I strongly feel that readers would benefit from having the highlights presented in a more accessible way. I suggest including a synopsis/summary table, which includes:

1. Satellite datasets evaluated
2. Spatial and temporal resolution of each dataset (raw, and when averaged for this analysis)
3. Statement of which datasets were analyzed close to the CO, i.e. not just within the 50 and 100 km radius, and what "close" means (as it varies for each dataset)
4. Key results (as summarized in the conclusions)

Point 3 above would also address the general inconsistency in the paper around what "close" means with respect to the CO

Reply: Presenting applied sensors and specifications is a good idea. A new table was included that highlights 1) sea-ice parameters investigated in this study, 2) the respective sensor/satellite and data distributor, 3) spatial and temporal resolution, and 4) the different radii that we investigated. Furthermore, the terms (close, medium, far range) are defined which we use in the manuscript. However, we believe that listing the Key Results in this table would duplicate the Summary chapter. We have therefore not implemented the latter.

**Specific Comments**

Title: The title would be more descriptive if it were "The MOSAiC Drift **first phase**…"

Reply: We agree: We now make clear in the title that the manuscript is about the first phase (October to July) only.

P1L26: Climatological mean over what period?

Reply: We now define the reference period in the sentence before. Note that the reference period was missing in the conclusion and in a few other sentences too. We now ensure that the term "previous" and "climatological mean" is defined, when it is used in the text (e.g. past 14 years, period 2005/2006 - 2018/2019, etc.)

P1L32: "…divergence**/convergence**…" i.e. be explicit that convergence is included in their definition of divergence. I don't feel it's widely assumed that divergence, when negative, can be used interchangeably with convergence.

Reply: Thanks for your comment. We now state everywhere explicitly in the manuscript and the figures whether we mean divergence or divergence and convergence.

P1L33-34: You state that 5 km is "close" the CO and that 50 km represents the "wider surroundings", but then that 50 km is "near" to the CO. Maybe just state the distances, to avoid ambiguity on what is close/near and what is not.

Reply: We agree. We now just state the distances in the abstract.

P2L46: Could you comment on whether the chosen floe was representative of the pack ice in general, or was an anomalously large floe selected? That strikes me as a big floe, especially for the Laptev Sea, and that's an important factor when considering differences in e.g. ice drift compared with previous years.

Reply: The selected floe wasn't anomalously large. We found several floes of same size that did survive summer 2019. In the paper https://tc.copernicus.org/articles/14/2173/2020/tc-14-2173-2020.html we investigate the initial conditions at the start of the expedition.

With respect to the ice drift: More than 50 GPS trackers were installed on the surrounding floes. Some of the surrounding floes were larger, but some were smaller. However, until Fram Strait was reached, no significant differences in drift speed were observed.

P2L56: The wording here is confusing. It would make more sense just to say "… were able to follow the ice floe back to its place of origin".

Reply: Thanks, sentence was changed

Section 2: Throughout this section I would have liked references to the relevant figures for each paragraph (e.g. sea-ice tracking, sea-ice concentration etc.) I'd also suggest changing the order of your figures so they better fit the flow of the text and make it easier for the reader to familiarize themselves with the datasets before moving to the discussion.

Reply: In the first version, most of the figures listed in chapter 2 were put into a supplement. These are now part of the manuscript again, as the TC has a strict regulation regarding the sequence of figures and the supplement content.

We agree that Chapter 2 is now somewhat overloaded. However, we fear that providing references to figures from Chapter 3 makes the manuscript more difficult to read. We have therefore decided to leave the sequence as it is and hope that it is sufficient.

P3L106: "…drift of the CO **and other buoys**…"

Reply: Thanks, changed

P3L199: How was the 6.25 km dataset averaged to get conditions close to the CO?

Reply: The ice concentration dataset has a grid resolution of 6.25 km. The satellite footprint of AMSR-E is 5 km and the AMSR2 one 4 km in diameter. Thus, these are the smallest scales we can resolve. This information is added to the section.

P4L130: Have you validated that the multi-parameter approach is more accurate, or is this statement just based on what's expected? Please specify.

Reply: We describe the OEM ice concentration dataset in more detail now in this section. The ship-based observations confirm that the OEM dataset is in better agreement. That is discussed in the results section 3.3.

P4L157-158: On average how many CS2SMOS SIT observations were averaged? It must be quite a small number.

Reply: We added the information to the manuscript:

*"The number of CS2SMOS SIT observations selected may depends on the position of the CO relative to local grid cell coordinates. The number varies between 10 and 14 grid cells for the 50 km and between 47 and 52 for the 100 km search radius. However, we do not expect this variability to cause a selection bias due to the smoothness of the CS2SMOS SIT data."*

P5L169-170: Do you mean that snow depth can only be retrieved for MYI, or that it's only MYI that has these limitations?

Reply: Thank you for the comment. We clarified that in the manuscript: Snow depth over MYI is limited to March and April. Over first-year ice, snow depth can be retrieved November to April.

P5L178: State what "MOY" and "MOD29" stand for

Reply: The MOD/MYD was unnecessary and is removed.

 Do you only use daily lead data up to April because this dataset isn't available after melt onset? Please specify.

Reply: Data is available from NOV to APRIL. The daily lead data can only be derived for winter months as the retrieval relies on a significant surface temperature contrast between leads and sea ice. Text is added accordingly.

P5L186: Is 10 km the MODIS resolution, or the distance you chose from the CO? If not the resolution, please state the resolution.

Reply: The resolution of the product is 1 km. Previous statement was wrong and is removed.

P5L196: Why was the ship outside the satellite coverage?

Reply: Between January and March, Polarstern exceeded the northern limit of the satellite coverage which is at about 88 °N for Sentinel-1. Occasionally, there were Sentinel-1 scenes available covering Polarstern at the end of January, but there were large temporal gaps and not sufficient spatial overlap to calculate reliable sea ice drift.

P6L228: Maybe I'm missing something but winds at the N. Pole don't look westward to me. It would be great to make the blue arrows in Figure 4 much clearer, to see the drift during the respective month.

Reply: We have updated Figure 4; the arrows depicting the monthly drift should now be better visible. Regarding the wind directions, note that the figure shows wind anomalies whereas the drift path is the actual drift. Combined with the climatological winds, the anomalies indeed lead to prevailing westward winds during those months. We have added a note of caution to the text to prevent that confusion.

P7L277-280: I disagree with the way the ice concentration analysis is presented here, for 2 reasons. 1) From October to July the concentrations don't necessarily "agree well". It's true that the MOSAiC concentrations don't deviate significantly from the long-term mean, but the patterns aren't the same, and during MOSAiC the concentration is consistently higher. 2) The lower concentrations in March are around the same time as the warm air intrusions. So, the drop in concentration is an artefact in the data rather than a "true" drop in concentration. You do go on to mention this below, but it should be included here to avoid any confusion that you're talking about "true" concentration.

Reply: We agree that our formulation was misleading and we reformulated it:

*"Averaged from October to July the ice concentration along the MOSAiC drift trajectory agrees well with the long-term 2005/2006 to 2019/2020 average (both have a mean of 97%). However, on shorter time scales there are significant differences: During the first half of the drift (October until end of February) the MOSAiC ice concentration was with 99.5% about 1% higher than the long-term average (compare black with blue line), while during the second half (March until end of July), it was lower than during the long-term average and shows higher variability than the first half. High ice concentration like the 99.5% are not unusual (compare to the grey lines) and can be expected in winter in the Central Arctic (e.g., Kwok, 2002). The*

*second half with lower (actually false) ice concentration is more unusual and will be discussed further in the following."*

P8L305: Are the daily values calculated using a monthly moving average? If not, how do you achieve daily coverage from CS2 data?

Reply: The daily values are based on the all SIT retrievals along the ground track of CryoSat-2 on a particular day. The thickness retrieval in the CryoSat-2 data is done for each waveform that is available at a rate of 20Hz and with a spacing of approximately 300 m on the ground. Thickness data from individual radar waveforms has substantial noise, thus we only use spatial averages here. We achieve the almost daily coverage because MOSAiC was located very close to the maximum orbit density of CryoSat-2 at 88 degrees north. There are a few days without CryoSat-2 l2p data coverage within the 100 km search radius in the beginning and the end, where the Polarstern was at lower latitudes. We expanded the description of the l2p data in section 2.3 to better explain the properties of this data set.

P10L385: What do you mean by "conditionally"?

Reply: Word was replaced by "to some extent".

Section 3.7: When not explicitly discussing positive divergence, I suggest changing "divergence" to "divergence**/convergence**". I don't feel it's widely assumed that divergence, when negative, can be used interchangeably with convergence.

Reply: We have adapted the manuscript in the way you suggested and stated always whether we mean positive divergence or divergence and convergence.

**Technical Comments**

P1L30: "month" -> "months"

Reply: Thanks, changed

P2L78: "…CO prior **to** departure…"

Reply: Corrected

P3L104: "number" -> "assess/gauge/quantify"

Reply: Corrected

P4L152: Remove the duplicate "sea" before "sea-ice"

Reply: Corrected

P7L264: "westerly" -> "eastward"

Reply: Changed

P7L276: "…radii, **we** will limit…"

Reply: Changed